# Texture coarseness responsive neurons and their mapping in layer 2–3 of the rat barrel cortex in vivo

Liora Garion[1], Uri Dubin[1], Yoav Rubin[1], Mohamed Khateb[1], Yitzhak Schiller[1]*, Rony Azouz[2]*, Jackie Schiller[1]*

[1]Department of Physiology and Biophysics, The Rappaport Faculty of Medicine and Research Institute, Technion-Israel Institute of Technology, Haifa, Israel; [2]Department of Physiology, Ben-Gurion University of the Negev, Beer-Sheva, Israel

**Abstract** Texture discrimination is a fundamental function of somatosensory systems, yet the manner by which texture is coded and spatially represented in the barrel cortex are largely unknown. Using *in vivo* two-photon calcium imaging in the rat barrel cortex during artificial whisking against different surface coarseness or controlled passive whisker vibrations simulating different coarseness, we show that layer 2–3 neurons within barrel boundaries differentially respond to specific texture coarsenesses, while only a minority of neurons responded monotonically with increased or decreased surface coarseness. Neurons with similar preferred texture coarseness were spatially clustered. Multi-contact single unit recordings showed a vertical columnar organization of texture coarseness preference in layer 2–3. These findings indicate that layer 2–3 neurons perform high hierarchical processing of tactile information, with surface coarseness embodied by distinct neuronal subpopulations that are spatially mapped onto the barrel cortex.

**\*For correspondence:**
y_schiller@yahoo.com (YS);
razouz@bgu.ac.il (RA); jackie@tx.technion.ac.il (JS)

**Competing interests:** The authors declare that no competing interests exist.

**Reviewing editor**: Michael Häusser, University College London, United Kingdom

## Introduction

The somatosensory system and especially the highly developed mystacial whisker system is one of the most important senses used by rats to sense the external world. The rat mystacial pad contains an array of vibrissae (whiskers) that rhythmically move back and forth (whisk) to palpate the environment (*Woolsey and Van der Loos, 1970*; *Carvell and Simons, 1990*). Using their whiskers, rodents can locate and distinguish objects in their immediate sensory environment (*Carvell and Simons, 1990*; *von Heimendahl et al., 2007*; *Diamond et al., 2008b*). Texture discrimination is one of the major sensory tasks performed by the barrel whisker system, and just a few whisker palpations are sufficient for discriminating between different texture coarseness with a high degree of sensitivity and reliability (*Guic-Robles et al., 1989*; *Carvell and Simons, 1990*). Previous studies have shown that activation of the primary barrel cortex is crucial for texture coarseness discrimination (*Guic-Robles et al., 1989*; *Carvell and Simons, 1990*; *Guic-Robles et al., 1992*), yet despite its importance the functional organization and texture coarseness coding in the S1 barrel cortex are largely unknown.

Previous studies of texture coding in anaesthetized rats and texture discrimination in awake behaving rats revealed that the degree of texture coarseness was correlated with the average firing rate of the granular and infragranular neurons (*Arabzadeh et al., 2005, 2006*; *von Heimendahl et al., 2007*; *Diamond et al., 2008a*; *Jadhav et al., 2009*). However, a simple averaged firing coding scheme was not sufficient to explain the psychophysical discrimination curve of the rats (*Arabzadeh et al., 2006*). Thus, other or additional coding schemes are required to achieve the high performances of the rat whisker system (*Jadhav et al., 2009*; *Morita et al., 2011*). One such additional coding scheme that

**eLife digest** As nocturnal tunnel-dwelling animals, rats rely on their whiskers to enable them to navigate in the dark. By moving their whiskers back and forth in movements called whisking, rats can obtain detailed information about the shape, texture, and size of objects in their path and also work out whether they can fit through narrow gaps.

Like all hairs, whiskers are made of dead cells. However, the follicles from which whiskers grow are densely packed with the ends of sensory nerves. When a whisker bends, information is transmitted along these nerves to a region of the brain called the barrel cortex, which takes its name from the barrel-shaped structures that represent the individual whiskers. These structures are arranged in an orderly grid with adjacent barrels corresponding to neighboring whiskers.

One of the key functions of whiskers is to help animals to distinguish between textures. By recording nerve impulses from the barrel cortex of anesthetized rats as the animals' whiskers were moved back and forth across four sandpapers of differing coarseness, Garion et al. have obtained key insights into how this process works. Only a small minority of barrel cortex neurons were more active or less active in response to an increase or a decrease in coarseness. Instead, most neurons showed a preference for one particular coarseness and fired much more when the rats' whiskers encountered the corresponding sandpaper.

Neurons with the same preferred coarseness were usually located close to one another. This gives rise to a gradient across the barrel cortex in which successive columns of neurons fired when the whiskers detected surfaces that were increasingly coarse, while neurons at different depths within each column fired at the same coarseness. This pattern was also seen using a different experimental method—in which whisker vibrations that were typical for different textures were converted to electrical signals and then played back to the rat's whiskers.

By revealing that the barrel cortex essentially contains a texture map, Garion et al. have identified a fundamental difference in how tactile (or touch) information is represented in the rat brain compared to visual and auditory information. Further work is now required to determine how the neurons identify different textures—whether they use particle size or sharpness, for example—and to confirm that the same results can be seen in awake animals.

has been suggested previously is a temporal code scheme of texture coarseness (*Arabzadeh et al., 2006*; *Wolfe et al., 2008*; *Jadhav et al., 2009*).

In different sensory modalities, sensory features are coded selectively by specialized neurons that respond best to certain parameters of the stimulus which in turn can be spatially mapped across the cortex. This spatio-functional representation is one of the fundamental principles governing organization of different sensory cortexes. For example, in the visual system, neurons that selectively respond to certain bar orientations or movement directions in the visual field are organized in columns (*Hubel and Wiesel, 1977*; *Ohki et al., 2005*). In the somatosensory whisker system the most conspicuous functional spatial map is the somatotopical organization of the vibrissae array onto the S1 cortex (*Woolsey and Van der Loos, 1970*; *Andermann and Moore, 2006*). Information about the organization within the barrel column is lacking. It has been shown that neurons in the barrel cortex selectively respond to certain angular directions of whisker deflection (*Bruno et al., 2003*; *Andermann and Moore, 2006*; *Lavzin et al., 2012*), which tend to be arranged in maps (*Hubel and Wiesel, 1977*; *Andermann and Moore, 2006*; *Lottem and Azouz, 2009*; *Kremer et al., 2011*). However, aside from angular tuning (*Bruno et al., 2003*; *Andermann and Moore, 2006*), mapping of other fundamental sensory features including texture coarseness onto the somatosensory barrel cortex is largely unknown. The present study sets out to use two-photon calcium imaging to further understand texture coarseness coding and explore the spatial representation of texture coarseness in layer 2–3 of the rat barrel cortex in vivo.

## Results

### Two-photon calcium imaging of layer 2–3 barrel cortex neurons during artificial whisking against textures

We recorded the responses of layer 2–3 neurons during artificial whisking (*Semba and Egger, 1986*; *Szwed et al., 2003*) ('Materials and methods') across sandpapers with different degrees of coarseness

(P120, P320, P600, and P1000) and whisking in air (free whisk–FW) while simultaneously recording calcium transients from multiple neurons bulk loaded with Fluo-4 using two-photon laser scanning microscopy (TPLSM) (*Figure 1A–D*). Sandpapers are traditionally used as a substrate of choice to investigate coarseness (*Guic-Robles et al., 1989*; *Diamond et al., 2008b*) as different sandpapers mainly differ by the average size of their particles (grit size), which are uniformly distributed across a flat surface. In experiments that included FW only three sandpapers were examined (P320, P600, and P1000) due to time limitation imposed by the dye washout observed in the AM bulk loading technique (*Sato et al., 2007*). To increase the temporal resolution during acquisition of calcium signals from large number of neurons, we developed a new free hand line scan routine that allowed us to retain high temporal resolution (50–100 Hz) while simultaneously recording from dozens of neurons (50.3 ± 3.4 neurons per experiment, range of 24–89 neurons per experiment, 23 rats) ('Materials and methods'; *Figure 1A*). The stability of the line scan path was checked continuously during the experiment (*Figure 1—figure supplements 1 and 2*). Responses were recorded from the principle barrel, as determined by intrinsic imaging during whisker deflection (*Figure 1—figure supplement 3*), and whisker displacements were measured by an infrared photo-sensor and a high speed camera (Flare, IO industries at 1000 fps) (*Figure 1B*, *Figure 1—figure supplement 4*; *Video 1*). Our imaging measurements showed that the calcium transients were locked to the whisking envelopes (*Figure 1B*). However, the temporal resolution of our calcium measurements (up to 100 Hz) was not sufficient to detect changes resulting from individual micro movements of the whisker. To characterize the variability and reliability of the calcium and whisking responses between repeated trails, we calculated the mean and SEM of the responses in individual neurons (individual traces are shown for four neurons in *Figure 1D* and the average values are shown for all neurons in a single experiment in *Figure 1E*). Furthermore, we tracked whisker movements during artificial whisking using the fast camera (1000 Hz) (*Video 1*). We found little variability in the whisker angle between consecutive trails and between different trial blocks (*Figure 1— figure supplement 4A*), indicating robust whisker movements during artificial whisking. As expected, we did observe differences in the whisker curvature (*Figure 1—figure supplement 4B*) and differences in the stick and slip events between different texture coarseness (*Figure 1—figure supplement 5*).

In our experiments, besides using intrinsic imaging to determine the principle barrel, the barrel boundaries and their relation to the bulk loaded neurons were more accurately determined using two additional methods. First, in all experiments, we identified the vertical blood vessels at the TPLSM plane and histological images, and vertically reconstructed down to the layer 4 barrel which was identified by cytochrome oxidase staining. Second, in some experiments (n = 7), we validated our conclusions by performing biocytin electroporation of selected bulk loaded neurons and later performed post hoc histology of cytochrome oxidase barrel staining aligned to the neuronal biocytin staining (*Figure 2*, *Figure 2—figure supplement 1*). For these experiments at the end of the imaging session several bulk loaded neurons were electroporated with biocytin using a patch electrode ('Materials and methods'). The biocytin-labeled neurons served as second clear anatomical landmark for alignment with the barrel boundaries (*Figure 2*, *Figure 2—figure supplement 1*). We found a good agreement between the two alignment methods in the seven experiments where both blood vessels and electroporated neurons were used.

Combined TPLSM and cell-attached patch clamp recordings from the same neurons (*Figure 1C*) indicated that TPLSM calcium imaging effectively detected most single action potentials (82% in all neurons and 91% in neurons with superior signal to noise ratio, see 'Materials and methods') and essentially all trains of two or more action potentials. In principle, the calcium responses could be presented either as the average *ΔF/F* or as peri-stimulus time histograms (PSTH) of spikes after transforming the calcium responses of individual traces to spike trains. We chose to present the data as averaged *ΔF/F* rather than spike trains for two main reasons: first, while our algorithm of transforming individual calcium responses to spike trains robustly detected firing events, determining the exact number of action potentials in each event was less reliable especially when firing is more intense as in our case. Second, reliable spike train transformation can only be performed in ~50% (56.4 ± 3.5%) of our recorded neurons in accordance with previous studies thus reducing the number of simultaneously analyzed cells (*Kerr et al., 2007*; *Sato et al., 2007*; *Rothschild et al., 2010*) ('Materials and methods').

## Texture coarseness preferring neurons in layer 2–3 of the barrel cortex

We compared the response of neurons to the different textures (sandpapers P120, P320, P600, P1000) and found that only a small minority of neurons showed either monotonically increasing or decreasing

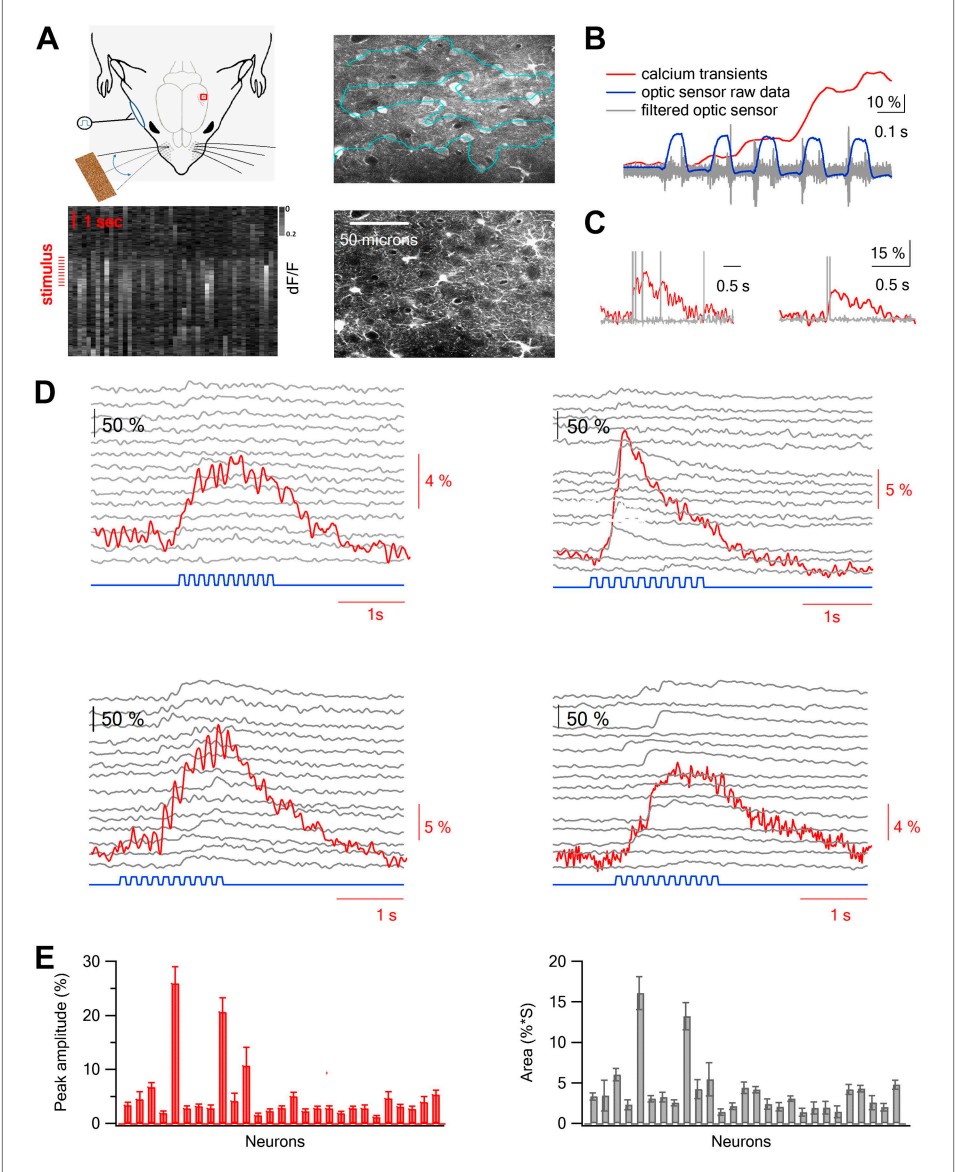

**Figure 1**. Two-photon calcium imaging of layer 2–3 neurons evoked by artificial whisking against textures in the barrel cortex in vivo. (**A**) Experimental set-up. A small (2 × 2 mm) craniotomy was performed above the barrel cortex, and the dura was carefully removed. Nerve stimulation was performed with a silver hook attached electrode to the buccal nerve. Only the principle whisker was left intact while other whiskers were trimmed. Artificial whisking (10 trains at 5.5 Hz) was performed either in free air (FW) or against different sandpapers. The calcium indicator Fluo-4 AM and the astrocytic dye SR101 were injected to layer 2–3 via a glass pipette under visual control using the two-photon microscope. The right upper panel shows an image of all cells filled with Fluo-4 and the right lower panel shows the astrocytes filled with SR101 in the same field of view (260 µm from pia). Left lower panel, calcium dependent fluorescent signals were collected from cells (horizontal aspect) using the free hand line scan mode (cyan line in right upper panel) and presented as a function of time (vertical aspect). Red lines mark the stimulus time. (**B**) Simultaneous recordings of the fluorescent calcium signal (red) and whisker trajectory (gray and blue) recorded using an optic sensor. The blue trace shows the unfiltered optical signal, and the gray trace shows the same optical data filtered (band pass of 50–500 Hz). 10% scale bar denotes the *ΔF/F* calcium transient signal. (**C**) Simultaneous voltage recording (gray) and calcium imaging (red) from a single neuron. The cell was bulk loaded with Fluo-4 and electrophysiological recordings were performed in the cell-attached recording mode from visually targeted neurons. Action potentials were partially truncated in the electrophysiological traces. (**D**) Example of the averaged calcium transient (red) and the corresponding single traces (14 repetitions, gray) from 4 single cells are

*Figure 1. Continued on next page*

*Figure 1. Continued*

presented together with the whisker stimulation (blue, 10 whisking cycles at 5.5 Hz). The traces in the different neurons were obtained simultaneously in the free hand line-scan imaging mode. (**E**) Average peak amplitude (±SEM) (red) and area (gray) of the calcium transients evoked by artificial whisking for all the cells (n = 51) recorded in one experiment.

The following figure supplements are available for figure 1:

**Figure supplement 1**. Free hand line scan position stability.

**Figure supplement 2**. Quantification of the free hand line scan position stability.

**Figure supplement 3**. Intrinsic optical imaging mapping of the principle barrel.

**Figure supplement 4**. Kinematic variables of a whisker movement in artificially whisking rat.

**Figure supplement 5**. Slip-stick events characteristic to different coarseness.

**Figure supplement 6**. The relative fluorescence contributed from neuropil and out of focus signals.

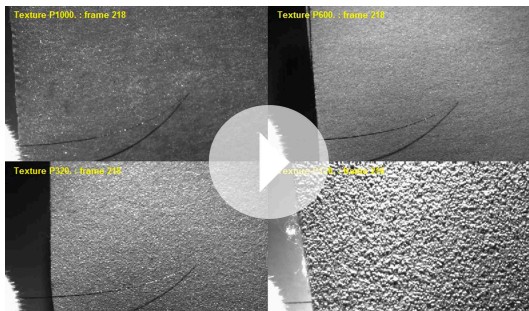

**Video 1**. Whisker movement against textures induced by facial nerve stimulation (artificial whisking). Whisking was induced by facial nerve stimulation (trains of 10 stimuli at 100 Hz repeated at 5.5 Hz). Whisker movement was captured with a fast camera (1000 Hz acquisition rate) while swiping 4 different textures (upper left P1000, upper right P600, lower left P320, lower right P120).

responses with surface coarseness (*Romo and Salinas, 2003*; *Arabzadeh et al., 2005*; *von Heimendahl et al., 2007*; *Diamond et al., 2008b*). Instead, many neurons responded preferentially to one of the four textures (*Figure 3A,B*). To quantify the response to different textures, we calculated the selectivity index (SI) for each neuron (see *Figure 3A,B* for examples; for definition of SI see 'Materials and methods'). We further defined neurons with S.I. ≥ 0.35 as texture coarseness preferring neurons (for more details see 'Materials and methods'). The SI calculations were performed independently using both the peak and area of the averaged calcium transients, and in 90.6 ± 3.2% of neurons, the preferred texture determined by the peak amplitudes and response areas corresponded.

Overall, texture coarseness preferring neurons were almost exclusively located within the principle barrel boundaries as determined with the post hoc histology with single cell resolution (*Figures 2 and 3C*). Aside from rare exceptions, neurons residing outside the barrel boundaries were either non-responsive or showed low selectivity (S.I. < 0.35) (*Figure 3C*). The average calcium transients of responsive cells in the septa regions was approximately 10% smaller than that recorded within barrel boundaries (3.7 ± 0.3% compared to 4.0 ± 0.2%, p < 0.01). Thus, it is unlikely that the lack of coarseness preferring neurons in the septa regions resulted solely from the small amplitude of the responses in the septa. On average 54.4 ± 3.5% of all recorded neurons within the barrel boundaries showed either texture coarseness or FW preference (*Figure 3C,D*). We further divided these neurons according to their preferred texture and found neurons that preferentially responded to each of the textures examined (*Figure 3D*). *Figure 3E* presents a cumulative histogram of the SI values of all responsive neurons.

To further confirm the TPLSM results, we performed electrophysiological extracellular single unit recordings from layer 2–3 neurons (293 neurons in 19 rats). In these experiments, we recorded the response to active whisking against five textures (P120, P320, P600, P1000, and compact disc [CD]). Similar to the TPLSM data, we found texture coarseness preferring neurons which maximally responded

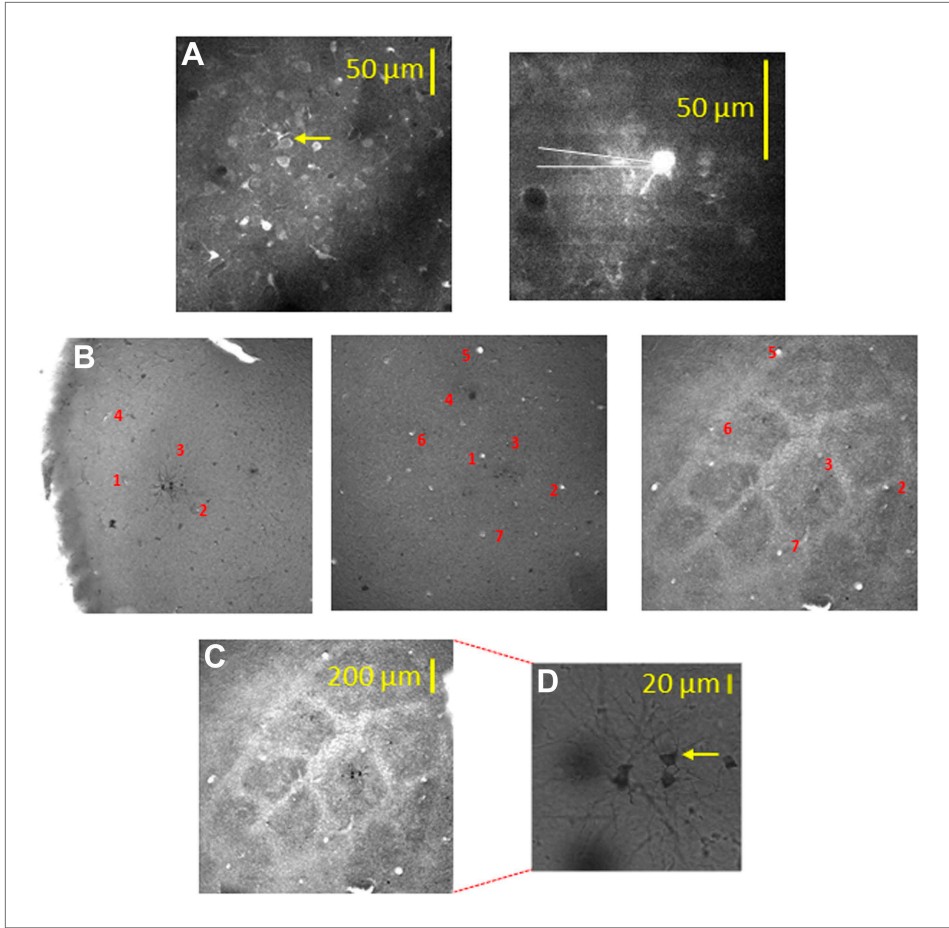

**Figure 2**. Alignment of the bulk loaded neurons in the TPLSM with the cytochrome oxidase barrel staining using biocytin electroporation of single neurons. (**A**) TPLSM Fluo-4 bulk loaded image of the imaged field. At the end of the recording session single cell electroporation of biocytin was performed on the bulk loaded neurons (typically, 1–4 calcium loaded neurons). Yellow arrow point to the one of the electroporated cells (right panel). These biocytin electroporated neurons served as a definite marker to the location of the calcium bulk staining. (**B**) Brains were fixed and sliced (100 μm thick) at the tangential plane, and barrels were visualized according to the cytochrome oxidase dense regions typical to layer 4 barrels. Alignment of the neurons labeled with biocytin (in cortical layer 2–3) to the layer 4 cytochrome oxidase stained barrel according to identified ascending blood vessels (designated by numbers 1–7) in three consecutive slices from layer 2–3 to layer 4 (slices: 100–200 μm; 200–300 μm; 400–500 μm). (**C**) Superposition of the biocytin-labeled neurons onto the identified barrel in layer 4 (C2 in this example). (**D**) The biocytin-labeled neurons are shown in higher magnification (arrow point to the identified cell in **A**).
The following figure supplement is available for figure 2:

**Figure supplement 1**. Alignment of the bulk loaded neurons onto the barrel using single elecrtoporated cells.

to each of the textures we examined (*Figure 4*). The percent of non-responsive neurons was lower in the electrophysiological recordings (2 ± 1.3%) (*Figure 4B*), probably due to the larger number of repetitions in the electrophysiological experiments. Taken together the electrophysiological findings further support our TPLSM findings with regard to the existence of texture coarseness preferring neurons in layer 2–3 of the barrel cortex.

## Response stability

An essential requirement for reliable comparison between the responses of layer 2–3 neurons to different textures is the stability of the responses evoked by the different textures during the experiments. To assess the stability of responses during our calcium imaging recordings, we divided the repetitions of each texture stimulus (typically ~30 repetitions) into two blocks segregated in time

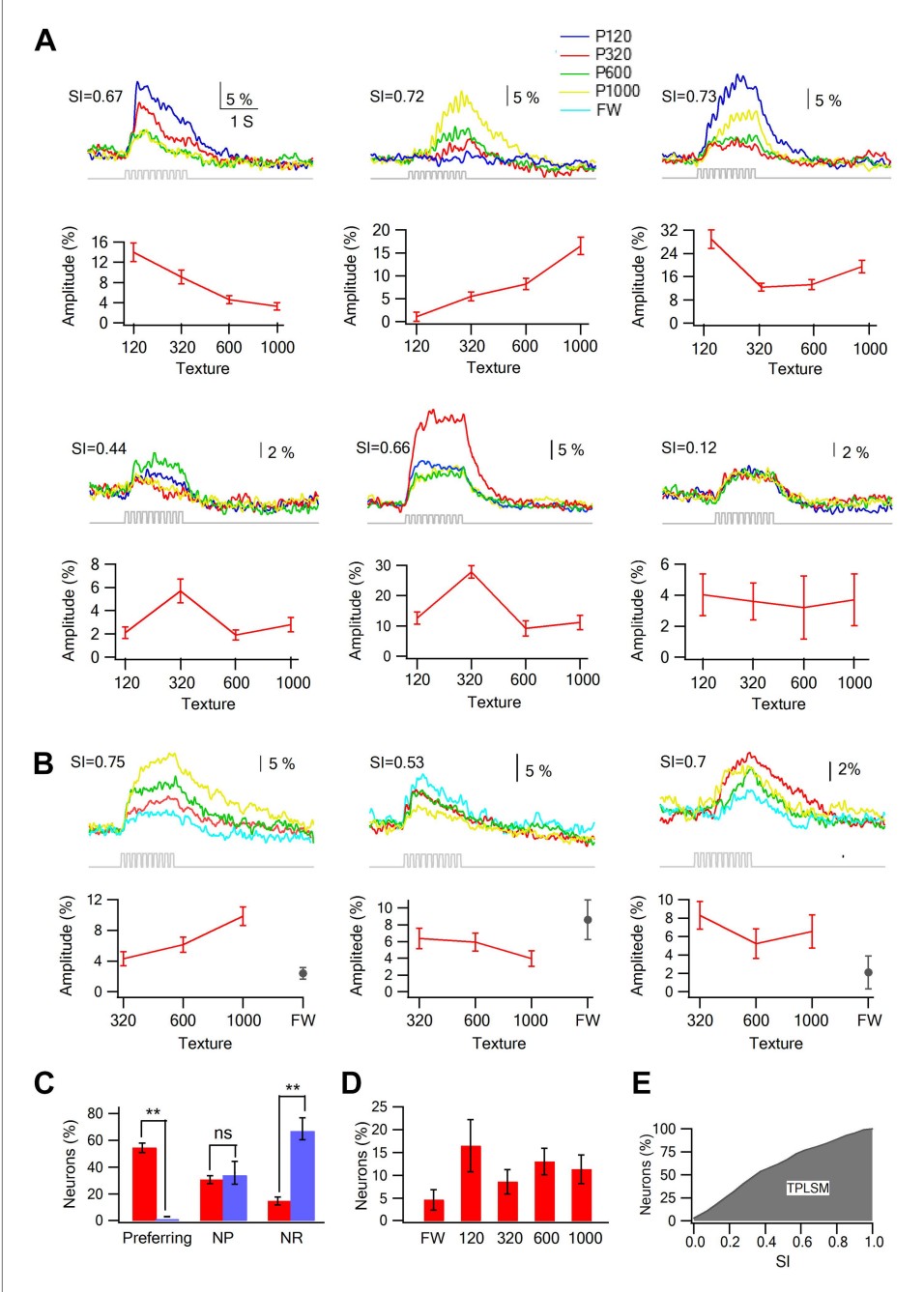

**Figure 3**. Texture preferring neurons in layer 2–3 barrel cortex evoked by artificial whisking: barrel vs septa regions. (**A**) Examples of the average calcium transient responses (30 repetitions) of six neurons to different textures (upper panels) and their corresponding tuning curves (lower panels; mean ± SEM) as calculated from the peak of the averaged calcium response evoked by each texture. For each cell the selectivity index (SI) is calculated and presented in the upper left corner. P120-blue, P320-red, P600-green, P1000-yellow. (**B**) Examples of the average calcium transient responses (30 repetitions) of three neurons to different textures and free whisking (FW) and their corresponding tuning curves (lower panels; mean ± SEM) as calculated from the peak of the averaged calcium response evoked by each texture or FW. For each cell the selectivity index (SI) is calculated and presented in the upper left corner. P320-red, P600-green, P1000-yellow, and FW-cyan. (**C**) The percentage of neurons (out of 1158 neurons within barrel boundaries in 23 experiments; 301 neurons in septa in five experiments) that were either coarseness preferring, non-preferring (NP) or non-responsive (NR). Neurons were divided to those within the barrel boundaries (red) and those located in the septa regions (blue). (**D**) The coarseness preferring neuronal population (S.I. ≥ 0.35) was further subdivided according to their texture coarseness preference (either P1000, P600, P320,

*Figure 3. Continued on next page*

*Figure 3. Continued*

P120, or FW). (**E**) Cumulative histogram of S.I. (in percent values) calculated from the calcium transient signals of all responsive neurons.

The following figure supplements are available for figure 3:

**Figure supplement 1**. Repeated stimulation sessions of same textures resulted in reliable calcium responses in layer 2–3 neurons.

**Figure supplement 2**. Stability of single neuron selectivity in repeated stimulation sessions.

and interleaved the different texture blocks ('Materials and methods'). In each experiment, we compared the peak and area of the averaged calcium transient responses of all recorded neurons (*Figure 3—figure supplement 1*) and of individual neurons (*Figure 3—figure supplement 2*) to each texture between the two blocks. Experiments were included in the analysis only if we did not

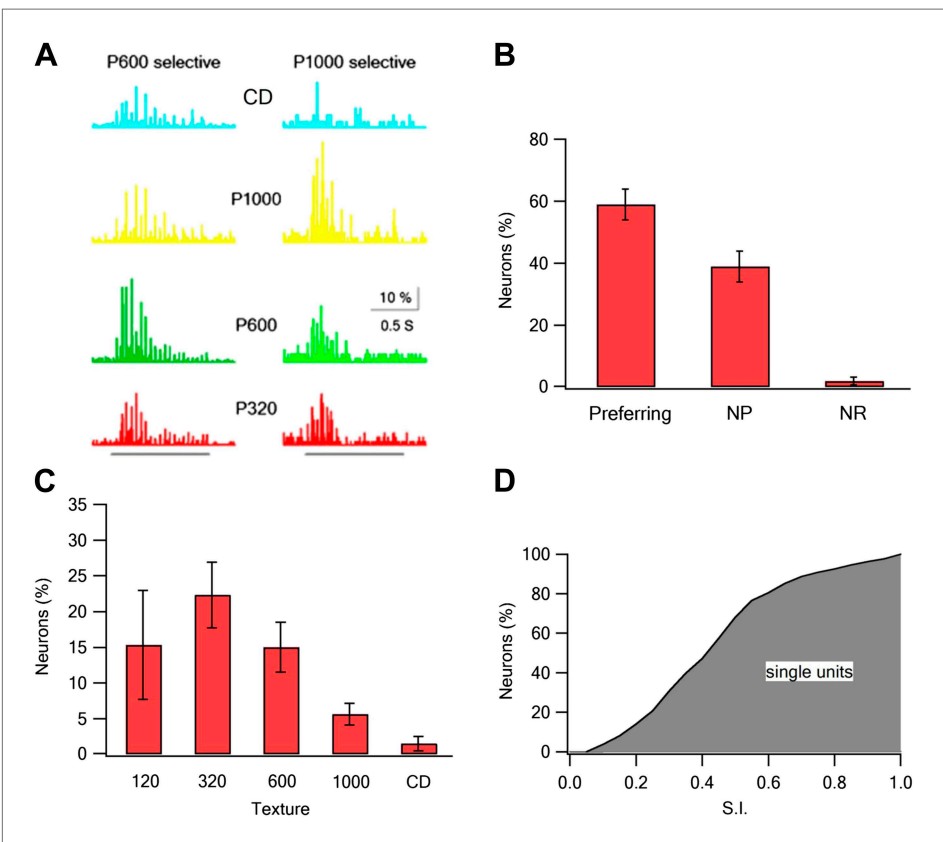

**Figure 4**. Coarseness preferring neurons as determined by extracellular single unit recordings. Multi electrode single unit recordings were performed using multi-contact extracellular electrodes from the S1 barrel cortex. (**A**) Example of PSTHs from two neurons in response to different texture coarseness (CD, P320, P600 P1000). Each PSTH was acquired during artificial whisking (10 whisks applied at 5.5 Hz) against CD and three textures (P320, P600, and P1000). The responses of 170 consecutive artificial whisk trains are summed and presented in 10-ms time bins. The underlying gray line designates the time of whisking train. Note that the two neurons prefer different coarse-nesses. The neuron presented on the right preferred the P1000 texture, while the neuron shown on the left prefer the P600 texture. (**B**) A summary of the responses acquired from single unit experiments (altogether 293 neurons from 19 rats). Responses were divided according to the percentage of neurons that were texture coarseness prefer-ring (S.I. ≥ 0.35), non-preferring (NP; S.I. < 0.35) or non-responsive (NR) to all textures examined. (**C**) The coarseness preferring neurons were further subdivided to neurons that preferred either P1000, P600, P320, P120 textures or CD. (**D**) Cumulative histogram of S.I. calculated from the single unit recordings of all responsive neurons neurons.

find a significant difference between the responses of the two blocks of all textures (*Figure 3—figure supplements 1 and 2*).

## Spatial maps of texture coarseness preference in layer 2–3 neurons during artificial whisking

We next investigated the spatial organization of neurons with respect to their preferred texture coarseness (*Figures 5 and 6*). The preferred texture coarseness is mapped for all responsive cells and for cells with selectivity indexes ≥0.35 (*Figure 5*). For both cases, we found spatial clustering of layer 2–3 neurons according to their preferred texture coarseness (*Figure 5*).

To quantify the degree of spatial clustering, we used two methods. First, we examined the probability of having the same preferred texture coarseness in the closest 1–4 neighboring neurons (*Figure 5B,C*). The preferred texture coarseness in the closest neighboring cell was 49.0 ± 3.9% compared to only 23.6 ± 3.4% expected from chance occurrence for cells with S.I ≥ 0.35 and 47.9 ± 13.2% compared to 36.6 ± 3.4% for all neurons (*Figure 5B,C*; $p < 0.01$). Significant differences between the measured and expected probabilities were also observed for 2–4 closest neighbors with the same preferred texture coarseness (*Figure 5B,C*). The second method we used to assess the clustering significance of neurons according to their preferred texture coarseness was the Monte Carlo analysis. Using this method, we compared the average distance between neurons with the same preferred texture coarseness in our experimental data with that of randomly distributed neurons (calculated for 1000 runs) (for details see 'Materials and methods' section). We found that in all our maps the average distance between neurons sharing the same texture coarseness preference was significantly smaller than that expected by chance occurrence (20 maps, at the $p < 0.05$ level, examples are shown in *Figure 5D*).

Aligning the TPLSM image with the histologically confirmed barrel boundaries (*Figure 2*, *Figure 2—figure supplement 1*) revealed that in all cases clusters of texture selective neurons were observed within the barrel boundaries, while septal neurons were mostly non-preferring and non-responsive neurons (*Figure 6*, *Figure 6—figure supplement 1*).

When we aligned multiple maps from different experiments and projected them onto a single 'normalized' barrel, we observed a tendency for texture coarseness to be arranged along the rostro/medial-caudo/lateral diagonal of the barrel with a tendency of the coarser texture (P120) to be represented in the rostro-medial region and the finer texture (P1000) at the caudo-lateral region (*Figure 6C*; $p < 0.01$ 4 way ANOVA). In addition, we found that the coarser textures tended to be preferentially represented at the perimeters of the barrel. Although the four-way ANOVA did not reach statistical significance, this was evident by the fact that neurons which preferred coarser textures (P120 and P320) had a significantly larger average radial distance compared with neurons that preferred the finer textures (P600 and P1000) ($p < 0.01$; *Figure 6C*). The statistical difference in the ANOVA test indicated that the arrangement of coarseness across the diagonal of the barrel is stronger compared to the radial arrangement.

To find out whether coarseness might be mapped according to the anterior-posterior positioning of the barrel within the barrel field as well, we plotted the dominant frequency preference of barrels as a function of their location along the row and arc of the barrel field. We did not observe any spatially consistent trend according to the arc or row barrel position (*Figure 6—figure supplement 2*).

## Columnar organization of texture coarseness preference along layer 2–3 of the barrel cortex

To investigate whether texture coarseness preference is retained along the vertical depths of layer 2–3 (z-axis), we performed single unit recordings with multi-contact, single shaft silicon probe electrodes. With these electrodes we were able to simultaneously record neurons from six different vertical depths across layer 2–3 (250–500 µm, inter-contact distance of 50 µm). We found that the probability of having the same texture coarseness preference in neurons recorded within all contacts of the same electrode was significantly higher than that expected from chance occurrence (3.5-fold for cells with S.I. > 0.34 and 2.2-fold for all cells; *Figure 7A–C*). We next calculated the probability of retaining the same texture coarseness preference across the z-axis depth of layer 2–3. For this analysis, we initially calculated the dominant preferred texture coarseness of neurons in the first contact (250 µm from the pia) and next calculated the probability that neurons in consecutive contacts will have the same preferred texture coarseness. The findings show that neurons tended to retain the same texture coarseness preference across the different contacts along the depth of layer 2–3 (*Figure 7C,D*).

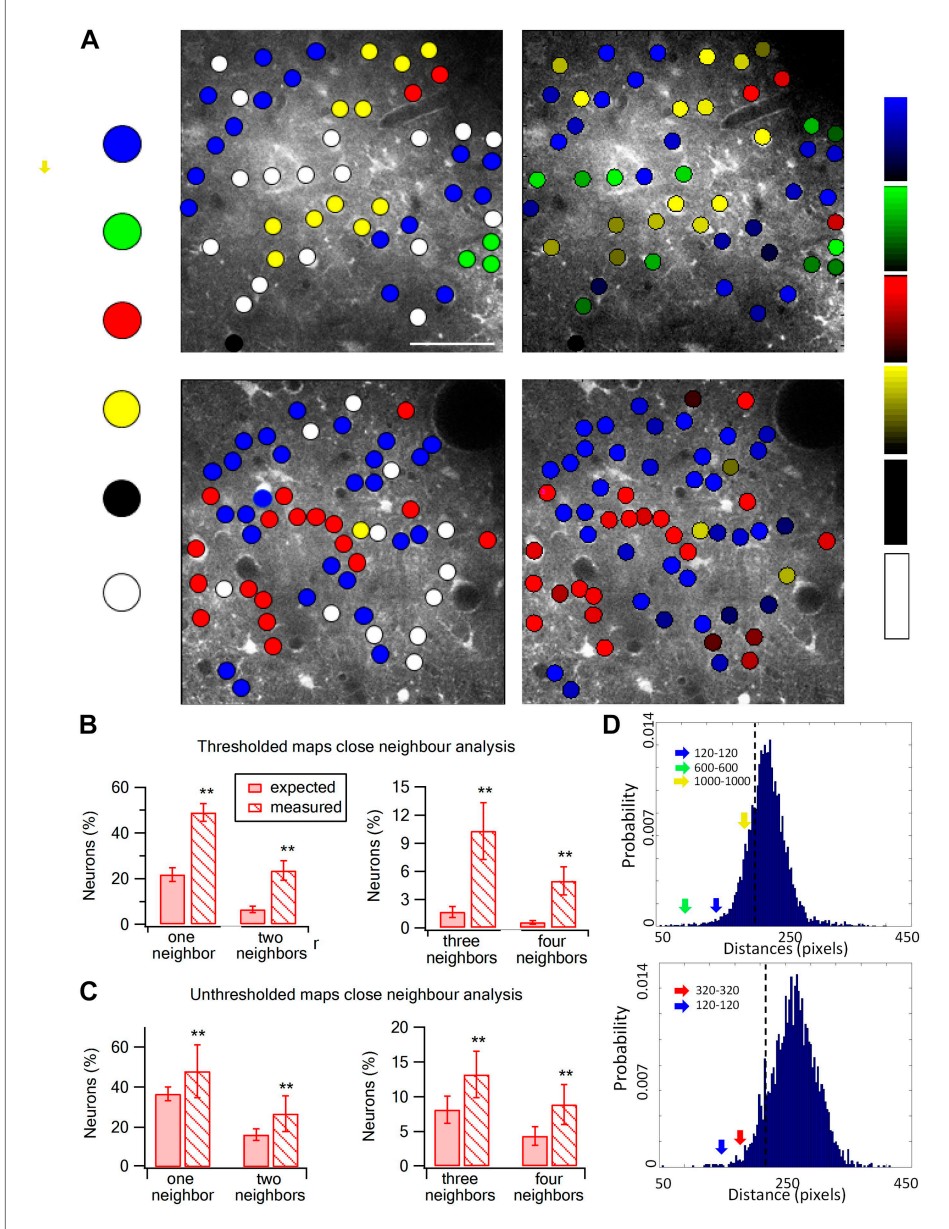

**Figure 5**. Spatial organization of texture preferring neurons during artificial whisking. (**A**) Texture coarseness preferring neurons from two experiments (upper and lower panels) were color coded according to their preferred texture (P120-blue, P320-red, P600-green, P1000-yellow, non-preferring-white and non-responsive-black). Left panels: only cells with S.I. ≥ 0.35 were included (S.I. thresholded maps). Right panels: all responsive neurons (unthresholded maps) from the same two experiments were color coded according to their preferred texture and the selectivity strength (value of the S.I. is scaled for each color and presented on the right ranging from 0 to 1). (**B** and **C**) The probability of having 1, 2, 3, or 4 next neighbors with similar preferred textures was calculated for the thresholded maps (**B**) and unthresholded maps (**C**) and compared with the probabilities expected from random spatial distribution (n = 12 rats) (\*\*p < 0.01). (**D**) Histograms of the Euclidian distance for 1000 runs for the two maps presented in **A**. To obtain these histograms the number of neurons that preferred each texture coarseness, as well as the number of non-responsive and non-preferring neurons was determined for each experiment. Later these neurons were randomly distributed in the recorded field 1000 times, and the Euclidian distance between neurons with the same preferred texture coarseness was measured. The broken line represents the lowest 5% of the histogram. The arrows represent the average experimental intra group distances for each texture coarseness (blue: P120-P120; red: P320-P320; green: P600-P600; yellow: P1000-P1000). Note that all intra-group distances in the two example experiments were below the 5% threshold value. The same result was obtained for all experiments tested (n = 20).

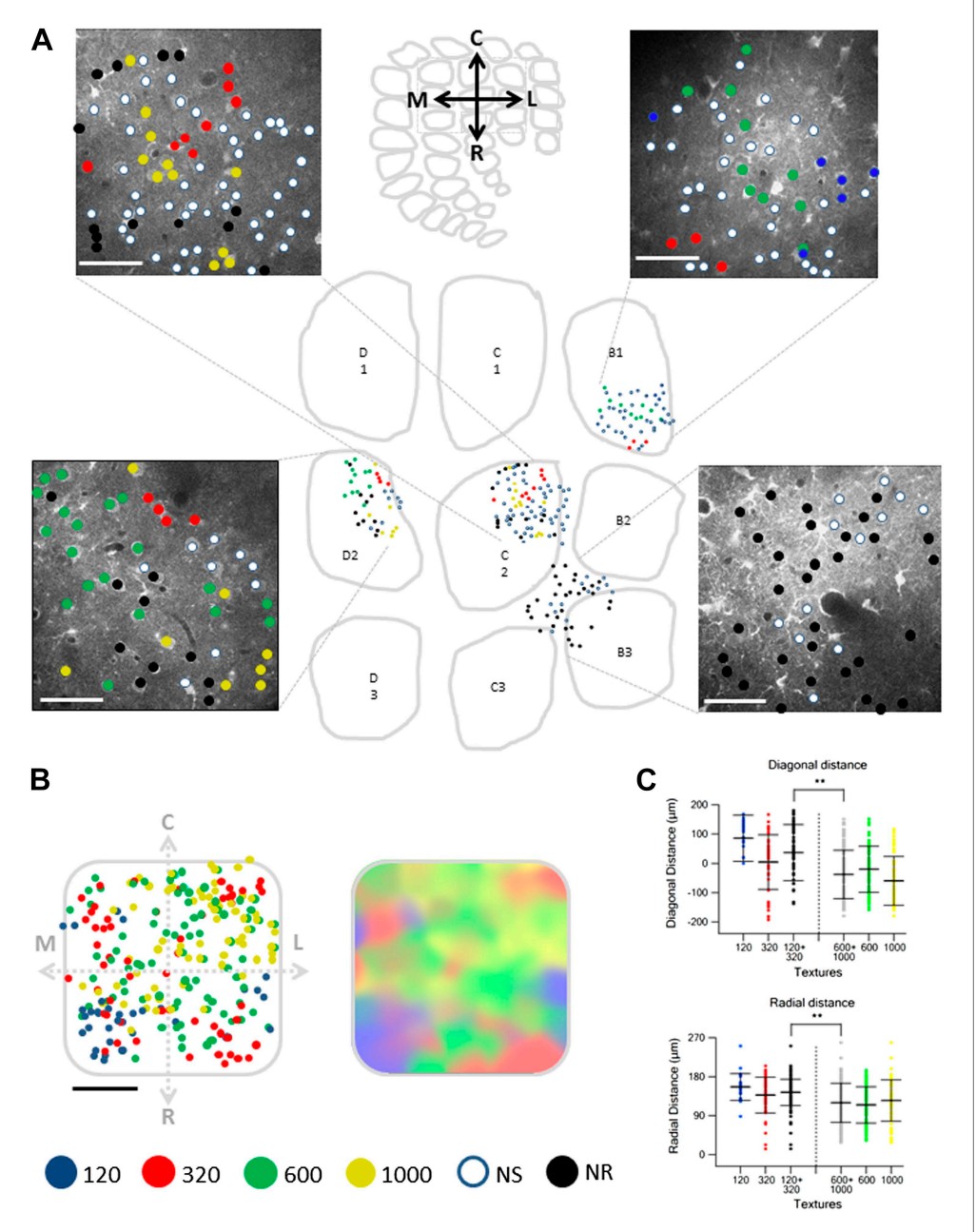

**Figure 6**. Spatial mapping of the preferred texture of neurons aligned to a normalized barrel. (**A**) Examples of four spatial texture preference maps taken from four different brains (3 inside the barrel boundaries and 1 in the septa) are presented. We carefully aligned the neurons with the barrel field (Scale bar: 50 µm) (see *Figure 2*; 'Materials and methods' for details). (**B**) Left: superimposition of the spatial maps from 12 experiments onto a 'normalized' barrel (Scale bar: 100 µm), middle panel smoothened map. In 7 out of the 12 maps imaging location was determined using the electroporation method as described in *Figure 2*; in the remaining maps we used vertical blood vessels alignment. (The preferred textures of neurons were color coded as P120-blue originated from 6 rats, P320-red originated from 10 rats, P600-green originated from 8 rats, P1000-yellow originated from 10 rats, non-preferring-white and non-responsive-black.) (**C**) Distance from the MC-LR diagonal (upper) and radial distance from the center of the barrel (lower) of each cell in the different experiments is presented for each texture separately and for the combined coarser textures (black, P120 and P320) and finer textures (gray, P600 and P1000) (**p < 0.01 for comparison of coarser and finer textures). Comparison of the four textures with ANOVA was not significant for the radial distance and yielded a p < 0.01 for the distance from the MC-LR diagonal.

*Figure 6. Continued on next page*

*Figure 6. Continued*

The following figure supplements are available for figure 6:

**Figure supplement 1**. Mapping of texture coarseness, non-selective and non-responsive cells relative to the barrel borders.

**Figure supplement 2**. The relationship between barrel location and dominant texture preference.

## Spatial maps of passive playback of texture-like whisker vibrations in layer 2–3 neurons

To better control for whisker movements and texture placement, we repeated the experiments using a fully controlled passive vibration protocol (*Figure 8*). With this protocol, we initially recorded the micro vibrations of the whiskers when contacting a rotating cylinder placed orthogonally to the whisker and covered with different sand paper textures (P100, P320, P600, and P1000). The actual micro vibrations of the whiskers when contacting the different sandpapers were measured by an infrared photosensor and converted to voltage signals. These voltage signals showed typical power spectrum characteristics for the different texture coarseness used (*Figure 8—figure supplement 1*). Later we used galvanometers to replay the typical micro vibrations evoked by the different texture coarsenesses to vibrate the principle whisker in anaesthetized rats.

Similar to the artificial whisking experiments, passively replaying the four different texture-like vibrational signals (corresponding to different sandpapers P100, P320, P600, and P1000) to the principle whisker resulted in a significant increase of firing to at least one of the simulated texture stimuli in 84.1 ± 13.4% of neurons within the barrel boundaries (*Figure 8*). Moreover, similar to the artificial whisking experiments, we have seen different response curves to the different texture-like vibrations (*Figure 8B*). 53% (52.7 ± 8.12%) of neurons within the barrel boundaries selectively preferred one of the texture-like vibrations (S.I. > 0.35; *Figure 8C*).

We next examined the spatial organization of texture-like preferring neurons. We found that similar to artificial whisking, neurons tended to spatially cluster across the S1 barrel cortex according to their preferred texture-like vibration (*Figure 9A*). We quantified the spatial clustering using the close neighbor analysis (*Figure 9B*) and found that on average the probability of having the same texture-like preference in the closest neighbor neuron was 40.9 ± 6.1% compared to 24.2 ± 4.0% as expected from random distribution (p < 0.01). Similarly, the probability of sharing the same texture-like preference with 2–4 of the closest neighbors was significantly higher than expected from random spatial distribution of neurons (*Figure 9B*).

Similar to the artificial whisking results, superposition of the passive maps onto a normalized barrel shows a strong arrangement along the rostro/medial-caudo/lateral diagonal of the barrel (*Figure 9D*). We also observed a tendency for the coarser-like textures to be preferentially represented at the perimeters of the barrel although this tendency was weaker compared to the active maps (*Figure 9D*).

Superposition of the combined passive and artificial whisking maps onto one normalized barrel further strengthen our conclusion with regard to spatial arrangement of texture coarseness along the rostro/medial-caudo/lateral diagonal of the barrel and to a lesser extent the radial arrangement of coarseness as well (*Figure 9—figure supplement 1*).

Taken together, the well-controlled passive replay of texture-like vibration confirm our findings with the artificial whisking paradigm and rule out the possibility that the spatial clustering of texture coarseness preference resulted from experimental variability in presenting the textures or other unknowns associated with artificial whisking.

## Discussion

In this study, we used two-photon calcium imaging to investigate the coding and spatial organization of texture coarseness in layer 2–3 of the S1 barrel cortex. We find texture responding neurons in layer 2–3 barrel cortex with subset of neurons preferentially responding to different texture coarseness. We further find that cells with similar preferred texture coarseness are spatially clustered across layer 2–3 of the barrel cortex. It is presently unclear whether these neurons code for the complex entity of coarseness or alternatively code for another more basic physical feature such as velocity of the whisker,

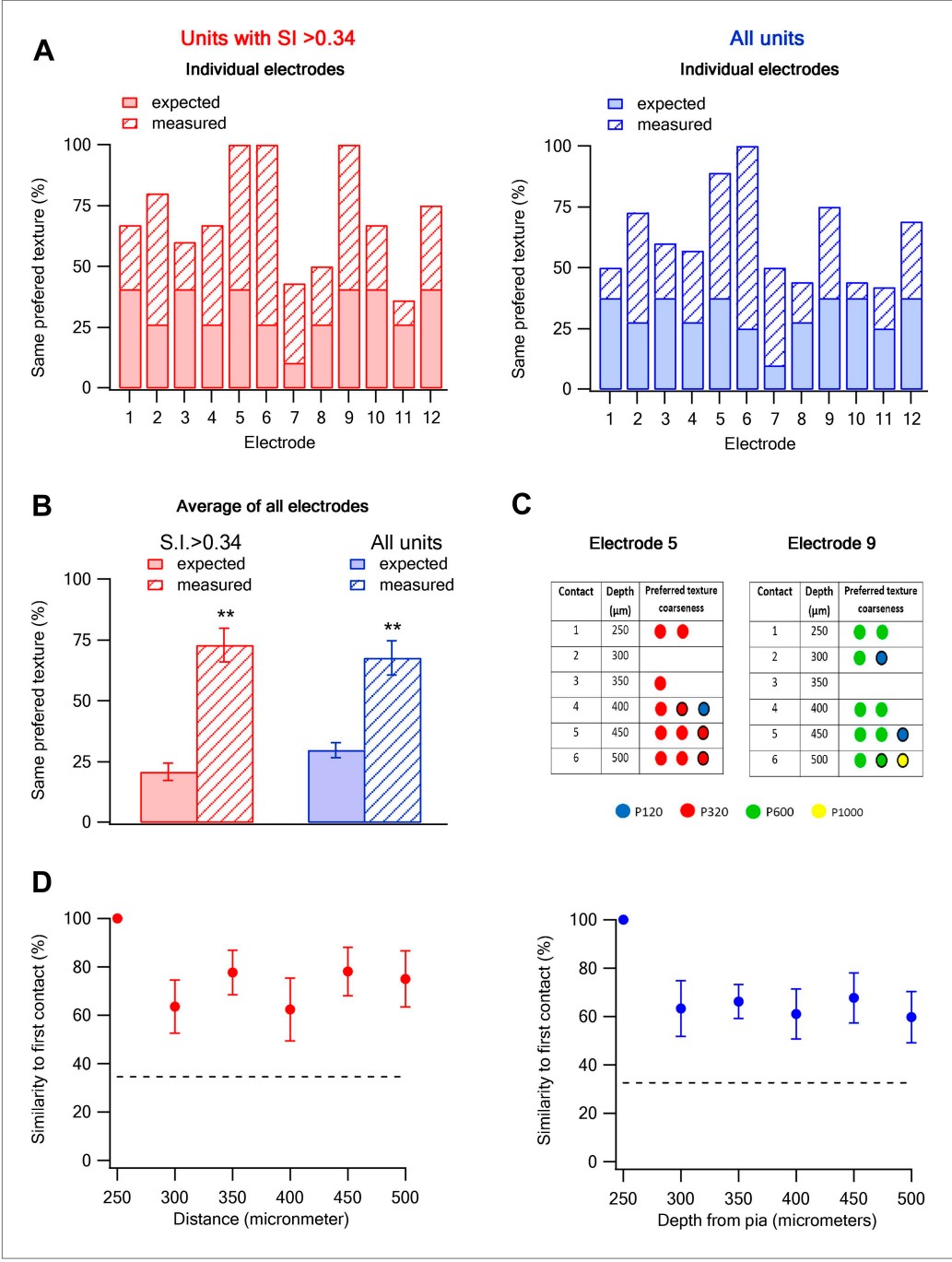

**Figure 7**. Columnar organization of preferred texture coarseness across layer 2–3 of the barrel cortex. (**A** and **B**) Comparison of the measured (solid) and expected (striped) probabilities for having the same preferred texture coarseness across all neurons recorded in different vertical depth of layer 2–3 using single shaft multi-contact electrodes (inter-contact distance of 50 μm). The data are presented for each individual electrode (**A**) and for the average of all electrodes (**B**). The data is also shown for all neurons with S.I. > 0.34 (red) and for all responsive neurons (average ± SEM) (blue). The expected probabilities were calculated from all 133 neurons in 12 electrodes from 12 rats (single multi-contact electrode per rat). (**C**) The preferred texture coarseness of all units across the six contacts of two example individual single shaft multi-contact electrodes from two different rats (electrode 5 and 9 in panel **A**) Circle with black perimeter mark units with S.I < 0.35. (**D**) The probability of retaining the same preferred texture along the different vertical contacts of a single electrode in layer 2–3. We initially determined the dominant preferred texture coarseness in the first contact (250 μm from the pia). Later, we presented the probability neurons

*Figure 7. Continued on next page*

*Figure 7. Continued*

will have the same preferred texture coarseness in consecutive contacts. For this analysis, we only used electrodes in which the first recording contact showed a clear dominant preferred texture (at least 66% of neurons with the same preferred texture, 9 out of 12 electrodes with 93 neurons).

grain sharpness, grain size, frequency etc. Regardless of the answer to this important question it is clear that texture coarseness can be decoded from the activity of these neurons irrespective of the exact physical entity they code for. These findings were shown in two different whisker activation paradigms, artificial whisking against different texture coarseness and passive replay of texture-like vibrations. The manner by which whiskers are activated is critical for the experiments. There is no 'perfect' way to repeatedly activate the whisker system in a robust and behaviorally relevant manner. Behaving rats performing texture discrimination tasks are of course most physiologically relevant, yet multiple well-controlled robust repetitions are very hard to obtain. On the other side of the spectrum passive whisker movements in anesthetized rodents are very well controlled and robust; yet probably differ from the physiological whisker activation. Artificial whisking in anesthetized animals is an intermediate whisker activation paradigm however receptor activation may deviate from that occurring under physiological whisking. It is important to note that a recent report shows that the kinematics of the whisker movement during artificial whisking closely simulates that of freely behaving rats (*Bagdasarian et al., 2013*). In our experiments both passive whisker activation and artificial whisking yielded similar results, reinforcing our main conclusion that texture selective neurons clustered across the S1 barrel cortex. It is however important to stress that in our experiments, rats were anesthetized; and in future, the findings should be corroborated in behaving animals. It is interesting to note that texture responding neurons were detected in barrel cortex of awake behaving mice during a texture discrimination paradigm (*Chen et al., 2013*).

## Texture coding in the barrel cortex

Using their whiskers, rats can reliably detect small differences in surface coarseness (*Guic-Robles et al., 1989*; *Carvell and Simons, 1990*). Psychophysical tests show that using their macro vibrissae, rats can discriminate between grain sizes as small as 10–20 μm (*Morita et al., 2011*). The underlying cortical coding mechanisms responsible for the formidable capability of rodents to distinguish texture coarseness have been studied extensively, mainly in granular and infra-granular cortical layers using multi- and single-unit recordings (*Guic-Robles et al., 1992*; *Arabzadeh et al., 2003*, *2006*; *von Heimendahl et al., 2007*; *Diamond et al., 2008a*; *Morita et al., 2011*). In the granular and infra-granular cortical layers of the barrel cortex neurons mostly respond to stick and slip events, and surface coarseness has been shown to be coded for the most part by the average spike count in the neuronal population. However, a rate coding spike count scheme could only explain discrimination between rough vs smooth textures (*von Heimendahl et al., 2007*), but could not explain discrimination between finer textures well within the psychophysical discrimination curve of the rat (*Arabzadeh et al., 2003*, *2005*; *Diamond et al., 2008a*). Thus, additional coding schemes based on the timing of action potential firing or on sparse synchronous firing have been suggested (*Arabzadeh et al., 2006*; *Diamond et al., 2008b*; *Jadhav et al., 2009*). In addition to the 'temporally based coding schemes', a 'resonance' coding theory has also been suggested. The resonance hypothesis suggests that texture identity is represented spatially across the whisker pad. This representation stems from the gradient of whiskers' lengths across the pad, denoting each whisker with a distinct resonance frequency (*Neimark et al., 2003*; *Andermann et al., 2004*; *Sato et al., 2007*; *Diamond et al., 2008a*).

Our results suggest a novel complementary strategy for coding texture coarseness in layer 2–3 of the barrel cortex. According to our results, coarseness is coded by neurons which prefer different coarsenesses, including the intermediate texture coarseness P600. We corroborated the existence of these neurons using electrophysiological single unit recordings. Interestingly, we found that neurons with similar coarseness preference are spatially clustered. Though certainly differences in coarseness induce differences in multiple parameters in whisker kinematics, the spatial clustering by itself may hint that coarseness is a fundamental quality that is represented in the rodent cortex. A coding scheme based on coarseness selective neurons can increase the reliability of texture coarseness coding as coarseness can be extracted by comparing the activity of neurons with different coarseness tuning.

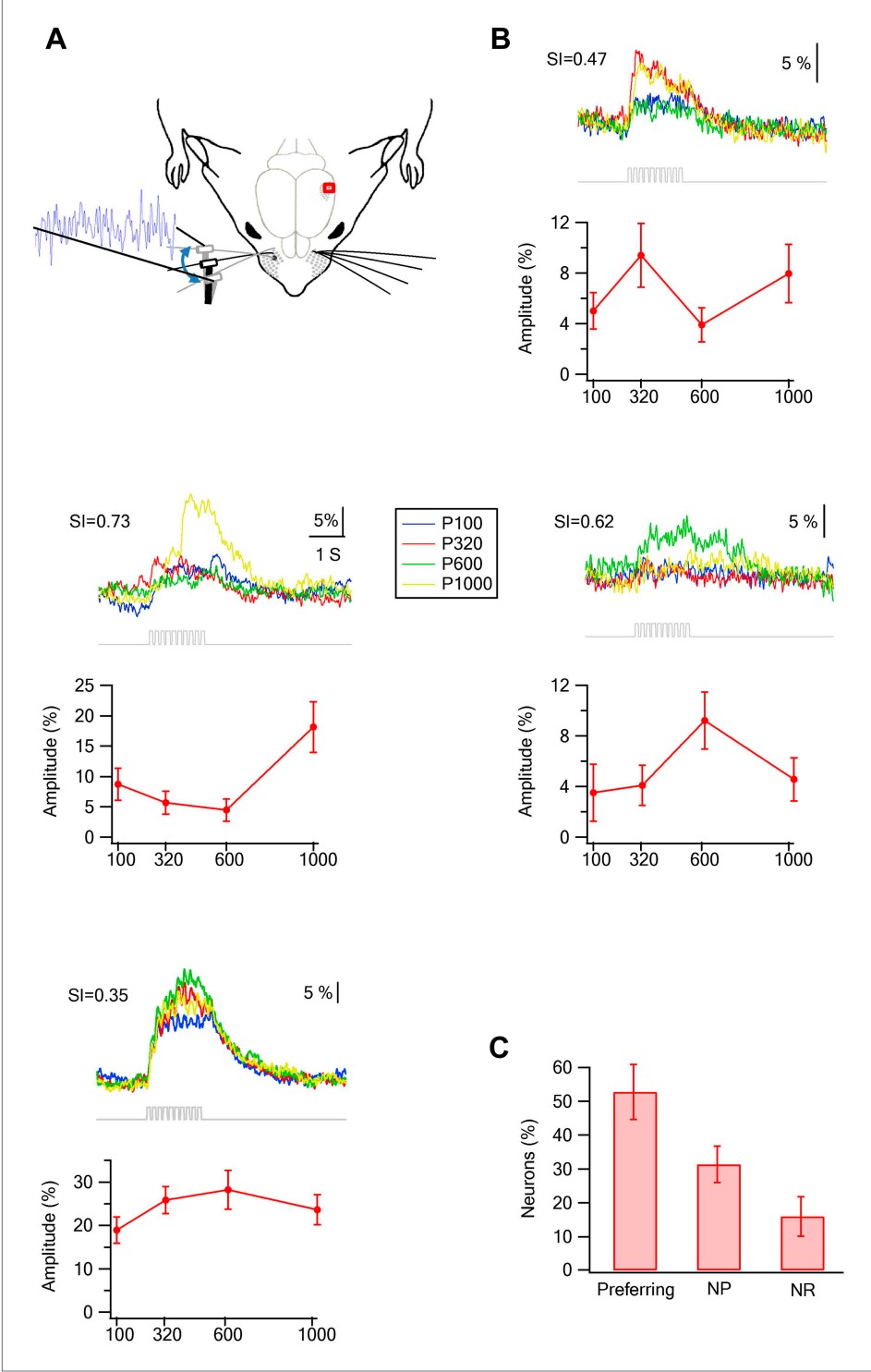

**Figure 8**. Preferred responses of neurons in layer 2–3 barrel cortex during passive replay of texture-like vibrations. (**A**) Experimental set up. The principle whisker was passively stimulated using a galvanometer with four texture-like micro vibrations at amplitudes of 15–30 µm. (**B**) Examples of the average calcium transient responses (30 repetitions) of four neurons to different passive coarseness-like vibrations simulating the four different textures (upper panels) and their corresponding tuning curves (lower panels; mean ± SEM) as calculated from the peak of the averaged calcium response. For each cell the selectivity index (S.I.) is calculated and presented in the upper left
*Figure 8. Continued on next page*

*Figure 8. Continued*

corner (P100-blue, P320-red, P600-green, P1000-yellow). (**C**) The percentage of neurons (out of 324 neurons in five experiments) that were either texture-like preferring, non-preferring or non-responsive.

The following figure supplement is available for figure 8:

**Figure supplement 1**. Passive whisker stimulation characteristics.

The alternative coding scheme where coarseness is extracted from the average firing of the neuronal population maybe more sensitive to the activity state of the animal and the context of the stimulus. This coding scheme is highly reminiscent of the coding suggested in primary visual and auditory cortexes in which subsets of neurons respond with increased firing selectively to a specific feature of the physical stimulus such as orientation or tone frequency (*Hubel and Wiesel, 1968*; *Nelken et al., 2004*).

As to the 'temporal coding' theories, the low acquisition rate of our calcium imaging data precluded us from critically testing these hypotheses, yet our findings do not contradict the use of temporal coding in the barrel cortex (*Arabzadeh et al., 2006*; *Diamond et al., 2008a*; *Jadhav et al., 2009*).

Most of the previous studies including this present study preferred to treat texture coarseness as a gestalt quality due to its complex nature and psychophysical importance. The downside of such an approach is that it does not address the question of what features are extracted by neurons during different coarseness representation. Keeping in mind that in sensory research one can only find tuning within the parameter space examined, designing the appropriate parameter space of stimuli that will encompass in unbiased way what features are extracted by neurons is a highly complicated issue and awaits further investigation. It is important to stress that this problem is generic to all sensory systems. Just as in the visual system, for example, both simple, artificial stimuli (e.g., drifting gratings), and more naturalistic stimuli (e.g., natural image patches) revealed complementary aspects of sensory coding (*Felsen et al., 2005*; *Gollisch and Meister, 2008*), also in the somatosensory system both simple passive and more complex natural stimuli are needed to reveal the full coding scheme of the system.

## Functional maps in sensory cortexes

The observation that functional information is mapped onto the cortical layers is one of the fundamental organizational principles of the central nervous system. In different sensory modalities, neurons which process closely related sensory information are contiguous. Examples of this basic organizational principle include orientation selectivity maps in the visual cortex and sound frequencies that are mapped in a tonotopic manner onto the auditory cortex of cats and monkeys (*Hubel and Wiesel, 1968*; *Stiebler et al., 1997*). Interestingly, in rats visual object orientation and auditory tonotopic cortical mapping were significantly less organized than in primates (*Ohki et al., 2005*; *Rothschild et al., 2010*). In layer 2–3 of the rodent visual cortex the neurons did not show the typical columnar organization rather a 'salt and pepper' organization in which orientation selectivity was not spatially organized (*Ohki et al., 2005*). Similarly, in the auditory cortex of rodents the tonotopic organization was only present on large-scale mapping, but broke down at finer scales (*Bandyopadhyay et al., 2010*; *Rothschild et al., 2010*).

In our study, using TPLSM calcium imaging, we could map coarseness preference in a population of S1 layer 2–3 neurons. In contrast to the primary visual and auditory cortexes in rats, we found a new spatio-functional organization in which similar texture selective neuron cluster together. These clusters were arranged along the rostro/medial-caudo/lateral diagonal of the barrel and to a lesser extent showed a tendency for radial arrangement of coarseness as well. This coarseness map is the second map described in S1 barrels in addition to the whisker angular direction map (*Bruno et al., 2003*; *Andermann and Moore, 2006*; *Kremer et al., 2011*). It is not known whether these two spatial maps (angular tuning and coarseness) are independent from one another or represent different aspects of a mutual physical property.

## Barrel and septa coarseness representation

Using biocytine electroporation, we overlaid the calcium maps with the anatomical coordinates of the barrels with single cell accuracy. From this analysis, we found that selective neurons to texture coarseness were almost exclusively confined to the barrel boundaries, whereas neurons in the septa area were non-responsive or showed low preference to texture coarseness. Previous anatomical and

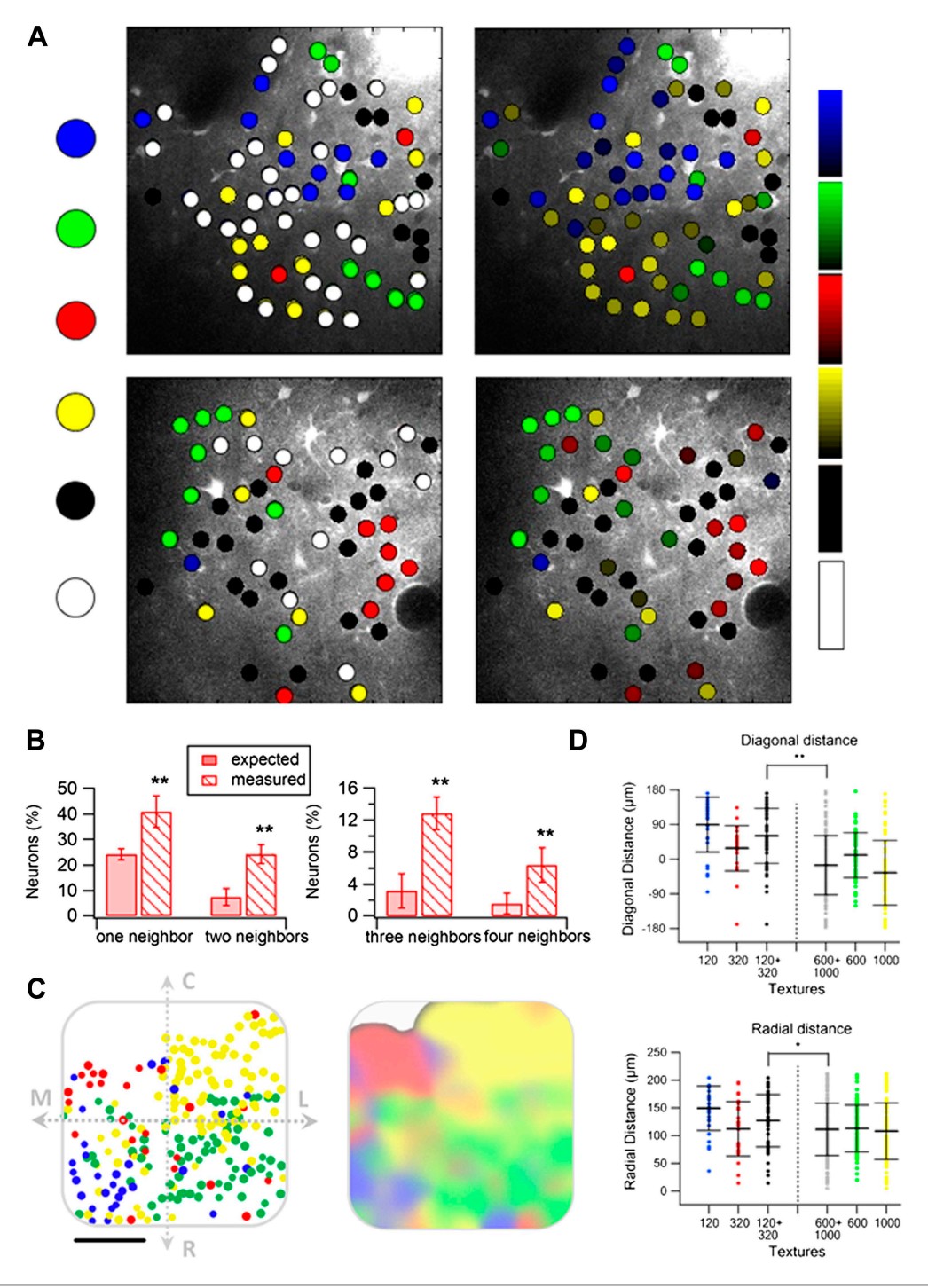

**Figure 9**. Spatial organization of responding neurons during passive replay of coarseness-like vibrational signals.
(**A**) (**B**) Left: texture-like preferring neurons (S.I. ≥ 0.35) from two experiments (upper and lower panels) were color
coded according to their preferred simulated coarseness. Right: all responsive neurons (with no S.I. thresholding)
were color coded according to their preferred texture coarseness (blue P120, red P320, green P600, yellow P1000,
white non-preferring neurons and black non-responsive neurons), and the selectivity strength (S.I. value) which is
depicted by the color scale (scale for each color is presented on the right, ranging from 0 to 1). (**B**) The probability
of having 1, 2, 3, or 4 next neighbors with similar preferred simulated coarseness was calculated for the unthresh-
olded maps and compared with the probabilities expected from random spatial distribution for all experiments
*Figure 9. Continued on next page*

*Figure 9. Continued*

performed (*n* = 6 rats) (**p < 0.01). (**C**) Superimposition of the passive maps from six experiments onto a 'normalized' barrel before (left), and after smoothing with a spatial filter (middle panel). In all six maps imaging location was determined using the electroporation method as described in *Figure 2*. (**D**) Distance from the MC-LR diagonal (upper) and radial distance from the center of the barrel (lower) of each cell in the different experiments is presented for each texture separately and for the combined coarser textures (black, P120 and P320) and finer textures (gray, P600 and P1000) (**p < 0.01 and *p < 0.05. Comparison of the four textures with ANOVA was not significant for the radial distance and yielded a p < 0.01 for the distance from the MC-LR diagonal.

The following figure supplement is available for figure 9:

**Figure supplement 1**. Spatial organization of responding neurons-combined artificial whisking and passive replay of coarseness-like vibrational signals.

functional studies have shown that barrel and septa neurons belong to different sub-networks associated with different thalamocortical and intra-cortical circuits (*Brecht and Sakmann, 2002*; *Shepherd and Svoboda, 2005*; *Bureau et al., 2006*; *Kerr et al., 2007*). The large differences in responses of barrel and septa neurons in our experiments are consistent with these findings. Our findings further suggest that texture coding is performed mostly by barrel neurons, yet these findings need to be confirmed in awake behaving animals.

## Layer 2–3 neurons as a higher order processing station

Taken together our findings in cortical layer 2–3 combined with previous findings from the lower cortical layers of the barrel cortex suggest that layer 2–3 neurons may represent a higher processing station compared with the granular layer 4 neurons (*Diamond et al., 2008a*). Our data show that layer 2–3 neurons, rather than using a simple rate coding scheme of the physical properties like layer 4 neurons (*Romo et al., 2003*), are organized in subpopulations of coarseness selective neurons, which are spatially mapped in the cortex. This coding scheme is consistent with the general strategy to extract fundamental features from the tactile stimuli and code them in spatially organized neuronal sub networks.

In this study, we did not investigate the mechanisms underlying the formation of coarseness clusters. We hypothesize that the emergence of coarseness clusters in layer 2–3 results from the feed forward connectivity patterns from lower cortical layers, possibly combined with higher connection probabilities between neurons belonging to the same coarseness cluster and dendritic amplification mechanisms (*Lavzin et al., 2012*).

## Materials and methods

### Surgical procedures

Wistar rats (male P27–35) were anesthetized with intra-peritoneal urethane (1.5 gr/kg body weight). After exposing the skull a stainless steel frame was attached to the bone using dental acrylic cement. A 2 × 2 mm wide craniotomy was opened above the barrel cortex and the dura was carefully removed. Brain pulsations were reduced by filling the craniotomy with 1.2% agarose in ACSF solution and covering with immobilized coverslip. The exposed cortex was super fused with normal rat ringer's solution. Body temperature was maintained at 36–37°C using a heating blanket (FHC). For the artificial whisking stimulation, the buccal branch of the facial nerve was exposed and cut at its proximal end (*Semba and Egger, 1986*).

### Intrinsic optical imaging

The principle whisker was identified using intrinsic optical imaging (*Grinvald et al., 1986*). Functional imaging was performed using a Qcam CCD camera (Q-imaging, Canada) equipped with a tandem lens system (DO-2595; F/0.95; and DO-5095; F/0.95; Navitar Inc., NY) and 610-nm LEDs (Telux VLWTG 9900; Vishay Electronic GmbH, Germany), while stimulating a single whisker by a galvanometer (Model 6210H, Cambridge instruments; 6 Hz deflections over 2 s duration) controlled via an isolated pulse stimulator (model 2100; A-M systems). The surface blood vessel pattern was imaged for reference. Image acquisition of the reflectance changes in the hemodynamic signal and analysis were made using a frame

grabber board (PCI-2110; National Instruments) and custom software written in our lab in the Matlab software. Images were acquired at 10 Hz frame rate (total of 200 frames per trial) with a 2 × 2 binning (~300 × 300 pixels, 7.4 μm pixel size).

## Fluorescence labeling and two-photon imaging

Bolus loading of the calcium indicator Fluo-4 AM (Invitrogen) to layer 2–3 neurons (250–350 μm below the pia) in the barrel cortex was performed as previously described (*Stosiek et al., 2003*; *Sato et al., 2007*). Briefly, the dye was first dissolved in DMSO (Sigma) and 20% (wt/vol) Pluronic acid F-127 to yield a concentration of 10 mM. This solution was further diluted to a final concentration of 1 mM in a solution composed of 125 mM NaCl, 5 mM KCl, 10 mM glucose, 10 mM HEPES, 2 mM $CaCl_2$, 2 mM $MgSO_4$, and 0.2 mM sulforhodamine 101 (SR101). The dye was injected into the cortex under visual control using a broken glass electrode (~10 μm tip diameter).

Two-photon imaging was performed using a Prairie two photon laser scanning microscope (TPLSM) platform (Prairie Technologies, Wisconsin, USA) equipped with a Ti: Saphire laser excitation (Spectra Physics) and a 40X water immersion objective lens (0.8 NA, Olympus). For calcium imaging, 870 nm excitation was used. Emission light from the Fluo-4 and the SR101 was collected simultaneously by two external photomultiplier (Hamamatsu; 570 nm dichroic mirror for separation of the emission to the two photomultipliers) and displayed using the software package Prairieview 3.1.0.0.

## Free hand line scan routine

A 'free hand line scan routine' was written in the lab (by Yoav Rubin) to allow faster control of the galvanometers and is now added to the Prairie View software. This enabled us to define a line that passed through large numbers of neurons with relatively high temporal resolution (acquisition rate of 50–100 Hz). To assure the stability of the line scan a reference image was captured every 3–5 scans and if needed corrections were made (*Figure 1—figure supplement 1*). Methods that allow for scanning along a user defined path were described in the past for a two-dimensional plane (*Lillis et al., 2008*) and for three-dimensional volume (*Gobel et al., 2007*). To obtain a free hand line scan routine, we created a scan path on a reference image which is built of a list of scanned pixel locations on that image. Later, we constructed a matrix of voltage values where Vx(i) and Vy(j) are the voltages that are needed by the X–Y galvanometer to drive the scanning beam to the ith and jth positions. Using this matrix, we can back-map the voltage values from a reference location on a scanned area.

A scan line is a vector or positions on the reference image defined as follows:

Line = {(Xi, Yi)|Xi is the X position of the (i)th pixel of the line, Yi is the Y position of (i)th pixel of the line, $1 \leq i \leq n$, n is the number of pixels along the scan line}.

With this scan mode the time resolution (scan time and the scan frequency) is calculated as:

Single-line-scan-period = n*dwellTime + MAX (X-return-time, Y-return-time), measured in seconds.

Scan frequency = 1/Single-line-scan-period, measured in Hz.

The return path is the straight line between the points (Xn, Yn) and (X1, Y1). X-return-time = X-delta/X-galvo-speed (measured in seconds); Y-return-time = Y-delta/Y-galvo-speed (measured in seconds). Where X-delta = |Xn−X1| (this is the distance between the scanline endpoints, along the X axis, measured in pixels).

Y-delta = |Yn−Y1| (this is the distance between the scanline endpoints, along the Y axis, measured in pixels).

dwellTime is the time it takes for the beam to scan a single pixel (this is configurable by the user).

Y-galvo-speed: the speed of the Y-galvo when not doing a scan (during the returning to the initial point movement), measured in pixel/second.

X-galvo-speed: the speed of the X-galvo when not doing a scan (during the returning to the initial point movement), measured in pixel/second.

This scan routine enabled us to define a line that passed through large numbers of neurons with relatively high temporal resolution, typically 50–100 Hz for more than 50 neurons.

## Single unit recordings and analysis

A multi-contact silicone electrode (NeuroNexus, Ann Arbor, Michigan) was inserted into the barrel cortex. The electrode was lowered using a precision stereotactic micromanipulator (TSE-systems, Germany). During the recordings the signals were amplified (X1000), filtered (0.1–10,000 KHz), and stored in a computer using the ME-16 amplifier and MC-Rack software (MEA, Germany). After completion of the experiments, the recorded data were replayed with a band-pass filter of 1–5 KHz to

obtain spikes. Single units were sorted with an offline spike sorter (OFS version 3; Plexon, Dallas Texas). The spike train data obtained after spike sorting were later analyzed using the NeuroExplorer (Nex Technologies, Littleton, Massachusetts) and homemade software using the MATLAB (MathWorks, NA) platform.

The single unit data are presented as peri stimulus time histograms (PSTH) with a 10-ms time bin. For determining selectivity and temporal dynamics of firing, spike counts were calculated along 100 ms time bins. The prolonged time bin was used for better comparison with the calcium imaging data which has prolonged time course. Selectivity calculations were performed on the spike count during the stimulation period. Selectivity to a texture or to CD was defined as showing both a p < 0.01 significance level in the ANOVA test when all textures were compared and a selectivity index of ≥0.35 (see data analysis for definition of selectivity index). Aside from rare cases, the preferred textures determined by the peak or area measurements coincided. These rare cases were excluded from the analysis.

## Cell attached patch clamp recordings from Fluo-4 labeled neurons

Targeted patch-clamp recordings were performed from neurons bulk loaded with Fluo-4 (n = 7) using patch pipettes electrodes (4–6 MΩ) (*Margrie et al., 2003*). Electrodes were filled with extracellular solution consisting of 135 mM NaCl, 5.4 Mm KCl, 1 mM $MgCl_2$, 1.8 mM $CaCl_2$, 5 mM HEPES, and 0.04 mM Alexa 594, pH 7.4. Gigaohm seal cell attached recordings were acquired at 10 KHz using a MultiClamp 700B amplifier, Digidata 1322A (Axon Instruments), and the acquisition software Clampex (Axon Instruments).

## Artificial whisker stimulation

Artificial whisking was induced by stimulating the buccolabialis motor branch of the facial nerve (*Brown and Waite, 1974*; *Semba and Egger, 1986*; *Szwed et al., 2003*). The nerve was cut, and its distal end mounted on a bipolar tungsten electrodes. Bipolar, rectangular electrical pulses (0.5–4.0 V, 40 µs duration) were applied through an isolated pulse stimulator (A360, WPI) at 100 Hz to produce whisker protraction, followed by a passive whisker retraction. The stimulation magnitude was adjusted to the minimal value that reliably generated the maximal possible movement amplitude. Typically, we evoked 10 consecutive trains (5.5 Hz, 50% duty cycle, 2 s) with 30 s between runs. All other whiskers except the principle whisker were cut off, the principle whisker was presented with different sandpapers (2 $cm^2$) with the tip of the vibrissa just touching the sand paper. The textures were manually placed to be coplanar with the trajectory of the whisker motion, and the distance from the whisker pad was set such that the whisker contacted the texture and made a full move on the texture without being stuck.

Whisker displacements were measured by an infrared photo-sensor (resolution: 1 µm; Panasonic: CNZ1120) placed 2 mm from the whisker pad. The voltage signals were digitized at 10 KHz and amplified or photographed using a high-speed camera (see next section). To control for stability of the responses for each texture stimulus, the repetitions (typically 30 repetitions) were divided to two separate blocks that were segregated in time and appeared in a random order during the experiment. Thus during the experiment, the different textures were randomly interleaved, and each texture was presented twice in two separate blocks (see *Figure 3—figure supplements 1 and 2*). Cases in which the same textures evoked significantly different responses (p < 0.01) in the two separate blocks were excluded from analysis.

Artificial whisking was performed against sandpapers of four different coarseness grades (the numbers in the parentheses indicate the average grain diameter): P120 (127 µm), P320 (46 µm), P600 (25 µm), and P1000 (18 µm). These coarsenesses were chosen both in accordance with previous studies (*Arabzadeh et al., 2005*; *Hipp et al., 2006*; *Lottem and Azouz, 2008*) and based on the findings that rats can discriminate between sandpapers varied by as little as 10–20 µm mean grain size (*Guic-Robles et al., 1989*; *Morita et al., 2011*).

## Whisker tracking

The whisking movement was photographed with a high speed camera (Flare, 4M180MCL, 4 Megapixel, Dalsa Xcelera-x4-CL, IO industries at 1000 fps) and software (Streams 6; IO industries) with resolutions 600 × 350 pixels. Movement of full-length whisker was tracked semi-manually, and the angle and curvature of the whisker were calculated as described in *Knutsen et al. (2005)* using a homemade software written in Matlab (MathWorks, NA). The mechanical parameters of the whisker were calculated from a spline function using cubic interpolation (Matlab *spline* function) that was fit to the whisker

tracking points in each frame. After interpolation, the 2D curve with coordinates $\{x(s), y(s)\}$ that matches whisker geometry is approximated by set of points $\{x_n, y_n\}$, where $n = 1\ldots N$. We compute first and second order discrete derivatives as

$$\dot{x}_n = x_n - x_{n-1}, \dot{y}_n = y_n - y_{n-1}, n = 2\ldots N,$$

$$\ddot{x}_n = \dot{x}_n - \dot{x}_{n-1}, \ddot{y}_n = \dot{y}_n - \dot{y}_{n-1}, n = 3\ldots N.$$

The whisker angle at its base was computed from the gradient of the whisker–spline curve as:

$$\theta(s)\big|_{s=0} = arctan\left(\dot{y}(s)/\dot{x}(s)\right).$$

In the discrete case, the angle at the base is given by:

$$\theta_{base} = arctan\left(\frac{\dot{y}_2}{\dot{x}_2}\right).$$

We approximate the standard formula of the 2D line curvature

$$\kappa_n = \frac{\left|\dot{x}\ddot{y} - \dot{y}\ddot{x}\right|}{\left(\dot{x}^2 + \dot{y}^2\right)^{3/2}},$$

using discrete curvature

$$\kappa_n = \frac{\left|\dot{x}_n\ddot{y}_n - \dot{y}_n\ddot{x}_n\right|}{\left(\dot{x}_n^2 + \dot{y}_n^2\right)^{3/2}}.$$

We computed global curvature $\kappa_{max}$ as the maximal value over all $\kappa_n$.

We designate $\theta_{base}[i]$, $\kappa_{max}[i]$ as base angle and max curvature at each image frame $i$.

The curvature and angle were found to be robust between the different repetitions (*Figure 1—figure supplement 4*; p < 0.01).

## Passive whisker stimulation

To passively move the whiskers, we placed a rotating cylinder covered with textures orthogonal to the whiskers. The cylinder (30 mm diameter) was driven by a DC motor (Farnell, Leeds, UK). We employed surfaces of four different grades (the numbers in the parentheses indicate the average grain diameter): P100 (162 µm), P320 (46 µm), P600 (26 µm), P1000 (18 µm). The cylinder surface was oriented so that the whisker rested on it and remained in contact during the entire session. For each texture, we recorded 50 revolutions (trials) per texture of the rotating cylinder, each lasting approximately 1 s. Whisker displacements transmitted to the receptors in the follicle were measured by an infrared photo-sensor (resolution: 1 µm; Panasonic: CNZ1120) placed 2 mm from the pad. The voltage signals were digitized at 10 KHz and amplified. The principle whisker was stimulated using a galvanometer (Model 6210H; Cambridge instruments). The voltages were delivered to evoke movements with amplitudes 15–30 µm (calibration of the galvanometer movement was performed using optical displacement measuring system resolution, µm; LD1605-2; Micro-Epsilon) (*Lottem and Azouz, 2009*). The whisker movements during passive whisker stimulation were characterized by their power spectrum (*Figure 8—figure supplement 1*; *Lottem and Azouz, 2009*). In order to measure spectral characteristics of this signal, we applied short time Fourier transform using pwelch in Matlab. Let $S_k(f_m)$ be Power Spectral Density (PSD) at frequency $f_m$ for texture $k$. Using power spectrum results, we estimated the center of mass (Power Centroid) for each texture

$$C_k = \frac{\sum\limits_{m=1}^{128} S_k(f_m) \cdot f_m}{\sum\limits_{m=1}^{128} S_k(f_m)}.$$

Thus, $C_k$ indicates frequency position of the PSD centroid for texture $k$.

## Histology

At the end of each experiment, a wide-field (500 μm × 500 μm) image stack of Fluo-4 cellular labeling was collected from the brain surface down to ~350 μm deep. Large radial vessels could be identified within these stacks. Then the neurons that were imaged with Fluo-4 were labeled with 5% biocytin using targeted electroporation via a glass pipette (10 V, square-wave pulses of 200 ms duration at 2 Hz for 5.5 s; *Pinault, 1996*). Thirty minutes later the animals were perfused with 0.1 PBS followed by 4% paraformaldehyde. The brains were removed and stored at the fixative for at least 24 hr. After fixation tangential slices (100 μm thick) were cut. Slices from cortical layer 4 (400–900 μm) were processed with the Cytochrome oxidase staining to reveal the barrel pattern. Slices from cortical layers 2–3 (0–400 μm) were processed with the 3,3-diaminobezidine (DAB) using avidin-biotin-peroxidase method for staining neurons filled with biocytin (*Horikawa and Armstrong, 1988*). All processed slices were mounted on slides embedded in Immu-mount. Imaging of the stained sections was conducted using a confocal microscope (Olympus FV 1000). Control of the microscope was done using Flouview10-ASW 2.1 software. The biocytin and Fluo-4 images were superimposed one upon the other with the neurons mutually stained in both images as anatomical landmarks. The relationship between the neurons (found in layers 2–3) and the barrels (stained in layer 4) was determined by the blood vasculature ascending through the cortex (*Figure 2*).

## Data analysis

For our analysis, we used custom-made analysis tools in Matlab (*Source code 1*; Mathworks, NA) and in Igor (Wavemetrics, USA) softwares. Relative florescence change $R_j[t_n]$ (ΔF/F) was calculated for each cell $j$ and each time sample $t_n$. Acquisition rate $Ts$ varied for individual line scans from 10 ms to 17 ms. Base line value $B_j$ was computed by averaging 10% of the pixels with minimal fluorescence $F_j[t_n]$. $R_j[t_n]$ was calculated using the following formula:

$$R_j[t_n] = \frac{F_j[t_n] - B_j}{B_j}.$$

The response of each neuron to the whisking stimulus was calculated by averaging the fluorescence signals of consecutive individual traces (typically 30 traces) obtained in response to the same texture.

To correct for baseline firing the baseline pre-stimulus value (1 s time window preceding the whisker stimulus) was fitted with a linear line and subtracted from the trace.

Responding neurons were defined as neurons with a peak average response that was significantly different than the pre-stimulus baseline value at the $p < 0.01$ significance level using the ANOVA test.

To determine texture selectivity, we calculated the selectivity index (SI) for each cell $j$. Texture selectivity was calculated in the following manner:

$$SI_j = \frac{Max(P_{jk}) - Min(P_{jk})}{Max(P_{jk})}.$$

When $Max(P_{jk})$ is the peak amplitude of the response to the best (preferred) texture ($k$) and $Min(P_{jk})$ is the peak amplitude to the texture with the smallest response. The peak response for each texture was determined by averaging 10 sampling points around the highest value of the response. In each cell, we also calculated the $SI$ using the area of responses (calculated during the whisking train). We applied this algorithm only to responding neurons. Neurons were considered selective if the $SI$ was larger than 0.35, and there was a significant difference (at the $p < 0.01$ significance level) between the responses to the different textures using the ANOVA (for equal variance, as determined by the Levene's test) or the Kruskal–Wallis statistical tests (unequal variance, for cases that failed the Levene's test).

With respect to the out of focus signal, this is an inherent problem of all imaging signals. Two-photon imaging minimizes this problem, but does not eliminate it all together. To partially deal with this problem we: (1) excluded all neurons in which the decay time constant was smaller than 650 ms (indicating a dominant neuropil component). (2) In the neurons in which we recorded simultaneously both electrophysiological and imaging data, we calculated the average transients with and without action potential firing. Traces without action potential firing represented the contribution of the neighboring out of focus neurons and neuropil signals. We found only very small transients (<10%) during non-responsive traces (*Figure 1—figure supplement 6*). (3) We compared the fluorescent values of

three structures in our images, cells, non-cell regions (neuropil and out of focus neighboring neurons), and blood vessels (corresponding to background fluorescence). The net fluorescent value in cells (after background subtraction, as measured in blood vessels) was 1845 ± 546 fluorescence units compared to 574 ± 376 fluorescence units and in non-cell structures (*Figure 1—figure supplement 6*), indicating a small out of focus and neuropil contribution in our recordings.

We used two methods to perform clustering analysis. First, we calculated expected vs measured closest neighbor probabilities. The expected closest neighbor probability for each experiment was calculated using the following equation where N depicts the number of cells:

$$P = \frac{N320}{(N120 + N320 + N600 + N1000)} \times \frac{N320}{Ntotal} + \frac{N600}{(N120 + N320 + N600 + N1000)}$$
$$\times \frac{N600}{Ntotal} + \frac{N1000}{(N120 + N320 + N600 + N1000)} \times \frac{N1000}{Ntotal}$$
$$+ \frac{N120}{(N120 + N320 + N600 + N1000)} \times \frac{N120}{Ntotal}.$$

The measured probability was calculated using

$$P = \frac{N\ same\ selectivity\ in\ closest\ neighbor}{(N120 + N320 + N600 + N1000)}.$$

Second, we have performed Monte Carlo analysis to identify the significance of the spatial clustering. For each pair of cells $j$, $k$ the Euclidian distance $d(X_j, X_k)$ between their position vectors $X_j$, $X_k$ is computed using $d(X_j, X_k) = \sqrt{(X_j - X_k)^T (X_j - X_k)}$. 2-D vectors $X_j$, $X_k$ describe cell position in image coordinates. The corresponding $NxN$ distance matrix $D_{jk} = \{d(X_j, X_k)\}$ is used to compute intra group (neurons with the same preferred coarseness) and extra group (neurons with different preferred coarseness) cell distances. For example, if a subset of cells $I_A = \{k: \text{cells } k \text{ prefer texture } A\}$. then between group $A$ and $B$ the distance is given by:

$$D_{AB} = \frac{1}{|A||B|} \sum_{j \in I_A} \sum_{k \in I_B} d(X_j, X_k),$$

|A| and |B|—are the number of cells in groups $A$ and $B$. The intra group distance is given by:

$$D_{AA} = \frac{1}{(|A| - 1)|A|} \sum_{j \in I_A} \sum_{k \in I_A} d(X_j, X_k),$$

where the term $(|A| - 1)|A|$ describes the number of elements without main diagonal.

In order to determine the significance of the clustering information, we compare the resulting distances $D_{AB}$ with randomly selected positions. Using the same number of cells, we randomly and uniformly selected the position of each cell in 2D image plane. Then the same distance criterion was applied. We computed a histogram of the between-cell distances for 1000 runs, normalized it, and selected the 5% population threshold value. This value indicates how the distances below this threshold differ from the uniform distribution. We found that the intra-group distances are significantly below the selected 5% threshold.

To generate smoothed texture coding maps of the normalized barrel each cell was assigned set of point coordinates $c^k(x_i, y_i)$ with $i = 1...N_k$, $N_k$—number of cells found for texture $k$, and $c^k$—color assigned to a texture $k$ and $x_i, y_i$ are the spatial coordinates of cell $i$.

For each texture, we generated a map of the normalized barrel. In order to fill the regions where there are no cells found we smooth each map by performing a convolution with 2-D Gaussian kernel which standard deviation of $\sigma = 15$ pixels. We get four maps $M^k(x, y)$ defined for all $x, y$ coordinates/pixels:

$$M^k(x, y) = \sum_{i=1}^{N_k} e^{-\left((x - x_i)^2 + (y - y_i)^2\right)/2\sigma^2}.$$

We normalize each map for each pixel (x, y) as follows

$$M^k_{norm}(x,y) = \frac{M^k(x,y)}{\varepsilon + \sum_{t=1}^{4} M^t(x,y)},$$

where $\varepsilon$ a small number that prevents division by zero. The physical meaning of this number is some cell response to a texture that we haven't measured.

Please note that $M^k_{norm}$ is scaled to range (0:1) and could be associated to the probability function.

We next generated an integrated colored map C(x, y) for all four textures, which represents a weighted mixture of the four individual color maps:

$$C(x,y) = \sum_{k=1}^{4} c^k M^k_{norm}(x,y).$$

For a very small value of $\sum_{k=1}^{4} M^k_{norm}(x,y)$, we set C(x, y) to be white.

Measurment of inter-trial spatial movement: We performed FFT-based correlation between two images, one is a target/template reference and the other is the current image data. Maximal value position in correlation image gives the relative shift between the two images in X and Y direction.

The template image was defined as a time average of the images in the selected trial:

$$I_{tmp}(x,y) = \frac{1}{N} \sum_{t=1..N} I(x,y,t).$$

where I(x, y, t) image for frame t from total N.

## Acknowledgements

We thank Irina Reiter for her technical help and especially for performing the histology. We thank Maria Lavzin for critically reading earlier version of the manuscript. This study was supported by the ISF and the Rappaport Foundation (JS).

## Additional information

### Funding

| Funder | Author |
| --- | --- |
| Israel Science Foundation | Jackie Schiller |
| Rappaport Foundation | Jackie Schiller |

The funders had no role in study design, data collection and interpretation, or the decision to submit the work for publication.

### Author contributions

LG, RA, JS, Conception and design, Acquisition of data, Analysis and interpretation of data, Drafting or revising the article; UD, Analysis and interpretation of data, Drafting or revising the article; YR, MK, Acquisition of data, Analysis and interpretation of data; YS, Conception and design, Analysis and interpretation of data, Drafting or revising the article

### Ethics

Animal experimentation: This study was performed in strict accordance with the recommendations in the Guide for the Care and Use of Laboratory Animals of the Technion Medical School. All of the animals were handled according to approved institutional animal care and use committee (IACUC) protocols of the Technion Israel. The protocol (IL-007-01-14) was approved by the Committee on the Ethics of Animal Experiments of the Technion. All surgery was performed under Urethane anesthesia, and every effort was made to minimize suffering.

## Additional files

### Supplementary file
• Source code 1.

---

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
