## [Decision Letter]

Thank you for resubmitting your work entitled “Texture Coarseness Responsive Neurons and their Mapping in Layer 2-3 of the Rat Barrel Cortex In-Vivo” for further consideration at *eLife.* Your revised article has been favorably evaluated by Eve Marder (Senior editor) and a member of the Board of Reviewing Editors. The manuscript has been improved but there are some remaining issues that need to be addressed before acceptance, as outlined below:

1) It is helpful that the authors show both smooth and unsmoothed barrel maps to compare their passive whisker stim paradigm with their artificial whisk paradigm, however they fail to report how coarseness of preferred texture varies with radial distance from barrel center (as they do in Figure 6). This is an essential quantification if they want to answer Reviewer 1's comment since the barrel maps themselves don't look that convincing. Also, in this section of the text they state that “the normalized map raises the possibility that textures coarseness is arranged along a diagonal axis”. However, if this was the case, one would expect there to be roughly equal numbers of very coarse neurons at the barrel edge as very fine neurons, with intermediate neurons clustered in the centre. This is inconsistent with the variation in coarseness preference with radial distance that they report in Figure 6. In fact, perhaps this diagonal mapping of texture is what causes the relationship between coarseness and radial distance in Figure 6 to be so weak since in order to calculate this they simply group the two coarsest textures into one “coarse” group and the two finest into one “fine group”. In order to reconcile the conclusion of Figure 9 with that of Figure 6 they should simply plot a scatter plot between coarseness and radial distance for both conditions. The shape of this relationship will show whether arrangement is concentric or diagonal within a barrel.

2) While the piezo map is an important addition to the paper, statistical testing of the map would be helpful. The artificial whisking map is reported as a radial map with coarse-selective neuron near the edges, the piezo data, however, does not look radial and is not tested with a similar statistical test as in Figure 6. Both maps appear to have a diagonal tuning from coarse in the medial/rostral corner to fine in the caudal/lateral corner. The maps should be confirmed with the same statistical test and compared against each other; for example a correlation analysis of diagonal position against texture tuning. It would also be interesting to superimpose the two maps and retest with more data points.

3) The addition of the extra panel in Figure 2 (right-hand panel) is helpful; however this figure is haphazardly laid out in general. It would not take much to scale all images and register them such that the narrative of Fluo4-imaging to cell localization is much clearer. Also if, as described in the rebuttal, they only did this in around 7/12 maps, and clearly obtain similar success in experiments without it, why bother at all? In summary: Figure 2 needs to be clearer, more streamlined and a justification given for why biocytin labelling was only needed for half the experiments.

4) The lack of response or texture tuning in septal neurons is intriguing! But it needs to be fleshed out a bit more in the main figures. If it is not possible to make extracellular recordings in septal regions, then it would be very nice to present septal imaging data as part of the population maps in the main figures. To confirm that the lack of response in septal neurons is not just the result of a few non-responsive experiments the authors could show examples of non-responding and responding cells on either side of a barrel edge from the same experiment.

5) The electroporation figure could do with some straightforward population analysis, for example location of electroporated cells superimposed on a normalized barrel.

6) The authors present images from one experiment to show cell position stability during imaging which is good to see, but it would be better to present a correlation analysis of cell position throughout the imaging procedure across a number of experiments to be sure that their imaging method is not subject to movement artefacts.

7) The analysis that was included in the rebuttal (demonstrating lack of tuning to coarseness across barrel field) should be included as a supplementary figure in the manuscript.

[Editors’ note: a previous version of this study was rejected after peer review, but the authors submitted for reconsideration. The previous decision letter after peer review is shown below.]

Thank you for choosing to send your work, “Texture coarseness responsive neurons and their mapping in layer 2-3 of the rat barrel cortex in-vivo”, for consideration at *eLife*. Your submission has been assessed by Eve Marder and two reviewers, in consultation with a member of the Board of Reviewing Editors. Although we find that your work is of interest, we regret to inform you that the findings at this stage are too preliminary for publication in *eLife*. In rare cases (such as this one) we think the potential interest of the work is such that we would welcome a new submission that would include additional experiments, although of course we cannot guarantee the results of its review.

As you will see from the reviewers' comments below, while both reviewers found your results interesting and potentially important for our understanding of texture discrimination in rodent barrel cortex, they also raised significant concerns about the strength of the conclusions that can be drawn from the existing results. We feel that significantly more data and more analysis are required in order to make the manuscript suitable for publication.

Reviewer 1:

In higher mammals, primary sensory cortical regions neurons are arranged into functional maps. Surprisingly, this has not been so clear at the cellular level in rodents. Here the authors describe a spatial clustering of cells with similar response properties to object texture in rat whisker barrel cortex using 2-photon calcium imaging. The imaging appears to be of good quality and the characterization of a functional map is important for the field. However, while the results are interesting I also find that there are several figures that should be improved with more N and/or more analysis to confirm the conclusions.

Major comments:

1) The electrical whisking paradigm attempts to move the whiskers in a manner more similar to the awake whisking rat. But, as discussed in the manuscript, it is difficult to have excellent control of whisker position and is not a perfect replication of awake whisking, which is why almost all researchers in the field use passive stimulation of whiskers when using anesthesia. The authors address this and show non-random spatial arrangements of cells in barrel cortex to passive stimulation. It would enhance the paper greatly to demonstrate whether the passively evoked responses also show functional maps when superimposed on a barrel as in Figure 6; more specifically: are passively stimulated coarse-selective neurons also located near the edges of the barrels?

2) The columnar organization of the preferred response in Figure 7 is a very interesting finding. However, it is unclear how many rats were used in these experiments, is it just one? These experiments should be repeated in different rats to increase N.

3) There is a clear and interesting difference of barrel and septal sensory responses, this should be followed up and confirmed with electrophysiological recording and identification of the recording sites.

4) The authors perform biocytin electroporation of imaged cells to locate the imaging site relative to the barrel. This is a nice technique and important for the conclusions of the paper but more analysis should be presented for a main figure. Is the yellow arrow in D an identified single electroporated cell? Are there confirmed neurons that have been imaged and stained: this should be presented? Was electroporation done in all experiments? Are there any axons visible going down through the barrel?

5) The free hand line scan method the authors use scans an optimal path through a number of neurons to provide a fast scan rate. The authors should describe how stable the optimal path is throughout their experiment. Small movements of the brain could end up in scanning neuropil. So, was the path corrected during recording to account for brain movement? A full frame image at the start and end of recording showing any brain movement would be a good control and should be presented.

Reviewer 2:

The authors examine whether an embodied feature of the natural stimulus space of vibrissal sensing, coarseness, demonstrates a spatial map. They provide compelling data that neighboring neurons are clustered in their response properties to variation in this dimension. I think that with further analyses to help us know how we should think about this finding the result is of significance.

My main questions were about what to make of this finding relative to features we know about and that we know have spatial maps; is this a re-discovery of direction tuning, because the frictional term of a given surface dictates how a vibrissa is stopped with high deceleration (e.g., caudal direction tuning?), or propensity to spring forward suddenly (e.g., rostral direction tuning?)? Similarly, while 'coarseness' has not been shown to have a map, frequency has been shown to have one related to biomechanical resonance, a property engaged by contact with sandpapers.

Analyses and clarifications conducted with the present data:

1) Did barrel identity in its anterior-posterior position in the barrel map predict the coarseness tuning of cells in that column?

2) When looking at the normalized barrel map, there seemed to be a lower left (presumably aligned to the somatotopic more ventral/posterior position on the face) to upper right gradient of frequency preference. For example, no cells preferring 120 were outside of the lower left, and the majority of cells preferring the highest frequency are upper right. Smoothing these data lightly and integrating across frequencies should be conducted.

3) Also, please plot the actual axes of orientation used to align the barrels on this summary image, and please make sure all maps are presented in this same orientation for consistency.

4) Is there a spatial radial or (as suggested in question 2) a lower left-upper right map for frequencies directly played into the vibrissae? These data should be plotted on a normalized barrel map also.

5) More generally, while I buy the argument that calcium signals cannot track individual stick-slip events, the mean frequency generated by artificial whisking on each sandpaper can be reported, as can the range and peak velocity values, as can the number of stick-slip events per grit. These properties for vibrissae chosen to be identity- and length-matched to the ones used for the initial data acquisition (as I assume those vibrissae are now long gone) should be used to make these measures. At least we can begin to try to understand what mapping may exist from lower level features that have been probed and the present findings.

6) The authors simply state that the contact point of the vibrissa was related to its 'not getting stuck'; can we have any more information? Was the contact point of the vibrissa on the surface of the sandpaper a typical distance from the face and/or from the vibrissa tip?

---

## [Author Response]

*1) It is helpful that the authors show both smooth and unsmoothed barrel maps to compare their passive whisker stim paradigm with their artificial whisk paradigm, however they fail to report how coarseness of preferred texture varies with radial distance from barrel center* (*as they do in*
Figure 6*). This is an essential quantification if they want to answer Reviewer 1's comment since the barrel maps themselves don't look that convincing. Also, in this section of the text they state that “the normalized map raises the possibility that textures coarseness is arranged along a diagonal axis”. However, if this was the case, one would expect there to be roughly equal numbers of very coarse neurons at the barrel edge as very fine neurons, with intermediate neurons clustered in the centre. This is inconsistent with the variation in coarseness preference with radial distance that they report in*
Figure 6*. In fact, perhaps this diagonal mapping of texture is what causes the relationship between coarseness and radial distance in*
Figure 6
*to be so weak since in order to calculate this they simply group the two coarsest textures into one “coarse” group and the two finest into one “fine group”. In order to reconcile the conclusion of*
Figure 9
*with that of*
Figure 6
*they should simply plot a scatter plot between coarseness and radial distance for both conditions. The shape of this relationship will show whether arrangement is concentric or diagonal within a barrel*.

Following the comments of the editors we now include the analysis of the radial distance for the passive stimulation paradigm as well (Figure 9). This analysis shows that similar to the artificial whisker paradigm, the combined coarser textures (P120+P320) had significantly larger radial distances compared to the combined smoother textures (P600+P1000), although the difference is less pronounced than that observed for artificial whisking.

Regarding the diagonal axis arrangement, we performed statistical analysis to investigate this point. For each cell we calculated the distances from the two diagonal lines. We found texture coarseness to be arranged along the medio/rostral-caudo/lateral diagonal of the barrel with a tendency of the coarser texture (P120) to be represented in the medio-rostral region and the finer texture (P1000) at the caudal-lateral region (Figure 6). In this case for both the active and passive maps 4 way ANOVA showed significance for comparison of the 4 texture.

Combining both these results, it seems that although there is a significant radial distances for coarser and smoother textures, the diagonal arrange is more prominent. This was especially true in the case of the passive case. These points are now communicated in the main text as well as in the figures of the revised manuscript (Figure 6, Figure 9 and Figure 9—figure supplement 1). These points are further discussed in reply to point 2.

*2) While the piezo map is an important addition to the paper, statistical testing of the map would be helpful. The artificial whisking map is reported as a radial map with coarse-selective neuron near the edges, the piezo data, however, does not look radial and is not tested with a similar statistical test as in*
Figure 6*. Both maps appear to have a diagonal tuning from coarse in the medial/rostral corner to fine in the caudal/lateral corner. The maps should be confirmed with the same statistical test and compared against each other; for example a correlation analysis of diagonal position against texture tuning. It would also be interesting to superimpose the two maps and retest with more data points*.

Following the editors’ comments we performed additional analysis and statistical testing:

A) We added the radial scatter plot for each of the 4 textures for the artificial whisking experiments (Figure 6).

B) We added the radial scatter plot for each of the 4 textures and for the coarser and finer texture for the passive whisking stimulation paradigm (Figure 9). We found that the passive experiments also show a statistically significant difference in the radial distance when combining finer versus coarser texture preferring neurons, although the difference was less pronounced compared to the artificial whisking experiments.

C) As suggested we added a combined map of the passive and artificial whisking data and compared the radial distance of neurons with different preferred textures (Figure 9—figure supplement 1).

D) We performed statistical analysis for the distance of cells from the diagonals. We found texture coarseness to be arranged along the medio/rostral-caudo/lateral diagonal of the barrel with a tendency of the coarser texture (P120) to be represented in the medio-rostral region and the finer texture (P1000) at the caudo-lateral region (Figure 6). This was true both when we performed a 4 way ANOVA and when we compared between coarser and finer textures for both the passive and artificial whisking maps. We added this analysis to Figures 6 and 9, and described the findings in the revised manuscript.

*3) The addition of the extra panel in*
Figure 2 (*right-hand panel) is helpful; however this figure is haphazardly laid out in general. It would not take much to scale all images and register them such that the narrative of Fluo4-imaging to cell localization is much clearer. Also if, as described in the rebuttal, they only did this in around 7/12 maps, and clearly obtain similar success in experiments without it, why bother at all? In summary:*
Figure 2
*needs to be clearer, more streamlined and a justification given for why biocytin labelling was only needed for half the experiments*.

As suggested by the reviewer we added a supplementary figure (Figure 2—figure supplement 1) in which we show the relative position of the electroporated cells, the TPLS imaged cells and the barrel border in all 7 experiments where we electroporated cells for determining barrel boundaries. It is important to stress that in all cases the electroporated neurons were also filled with the calcium sensitive dye, and thus could also be identified in the two photon image. In some cases electroporation intentionally targeted neurons slightly off the imaged field in cases where we observed de-staining and bleaching of the imaged neurons at the end of the experiment.

With respect to the number of experiments performed with biocytin electroporation. We routinely used 3 methods of localizing the barrel, intrinsic imaging, cytochrome oxidase staining and vertical blood vessels reconstruction to define the relationship between our layer 2-3 imaging field and the stained layer 4 barrel. The electroporation method was performed on only 7 experiments since it was added later in the project in an attempt to cross check and validate the two other methods. We clarified this point in the revised manuscript.

*4) The lack of response or texture tuning in septal neurons is intriguing! But it needs to be fleshed out a bit more in the main figures. If it is not possible to make extracellular recordings in septal regions, then it would be very nice to present septal imaging data as part of the population maps in the main figures. To confirm that the lack of response in septal neurons is not just the result of a few non-responsive experiments the authors could show examples of non-responding and responding cells on either side of a barrel edge from the same experiment*.

Following the editors’ comments we added information regarding septal neurons.

In the revised manuscript we have made the following changes:

A) In the revised manuscript Figure 6 panel B includes only the texture preferring neurons recorded within the barrel border. We preferred not to include the non-responsive and non-selective neurons in this main figure to make the presentation of texture preference mapping clearer. We felt that adding non-responsive and non-selective neurons to this figure may compromise the main point of the manuscript.

B) In Figure 6—figure supplement 1 we present the superimposed mapping of all neurons (including non-responsive and non-selective neurons) located within and out of barrel boundaries onto a “normalized” barrel (upper panel). In addition we show 3 maps of individual experiments (lower panels) to exemplify the point that responding cells were mostly located inside the barrel border while outside barrel borders cells are mostly non-responding or non-selective cells within the same map.

*5) The electroporation figure could do with some straightforward population analysis, for example location of electroporated cells superimposed on a normalized barrel*.

As suggested by the reviewer we now added new supplement figure (Figure 2—figure supplement 1), which includes all electroporated cells superimposed on a normalized barrel.

*6) The authors present images from one experiment to show cell position stability during imaging which is good to see, but it would be better to present a correlation analysis of cell position throughout the imaging procedure across a number of experiments to be sure that their imaging method is not subject to movement artefacts*.

Following the reviewers’ comment we present now a quantification of the cells movement across different experiments. More specifically we calculated the relative shift of the image in both the X and Y directions compared to a template picture during the different trials (described in the revised Methods section). In Figure 1—figure supplement 2 we present the trial-by-trial movements in the X and Y directions for one experiment, and the average movement per trial in X and Y directions for 15 experiments. We found very small shifts between the images throughout the experiment, on average per trial the X-Shift was 0.235±0.54 and Y-Shift was 0.215±0.58; n=15 experiments).

*7) The analysis that was included in the rebuttal* (*demonstrating lack of tuning to coarseness across barrel field) should be included as a supplementary figure in the manuscript*.

As suggested this figure is now added to the supplementary figures as Figure 6—figure supplement 2, and a description in the main text.

[Editors’ note: the author responses to the previous round of peer review follow.]

Reviewer 1:

*1) The electrical whisking paradigm attempts to move the whiskers in a manner more similar to the awake whisking rat. But, as discussed in the manuscript, it is difficult to have excellent control of whisker position and is not a perfect replication of awake whisking, which is why almost all researchers in the field use passive stimulation of whiskers when using anesthesia. The authors address this and show non-random spatial arrangements of cells in barrel cortex to passive stimulation. It would enhance the paper greatly to demonstrate whether the passively evoked responses also show functional maps when superimposed on a barrel as in*
Figure 6*; more specifically: are passively stimulated coarse-selective neurons also located near the edges of the barrels?*

Following the reviewer's comment we now present the passive maps superimposed onto a normalized barrel with and without smoothening (see Figure 9; Smoothing is described in the Methods section). Similar to the artificial whisking map, the normalized passive map also showed a tendency for coarser textures to map on the barrel perimeters.

*2) The columnar organization of the preferred response in*
Figure 7
*is a very interesting finding. However, it is unclear how many rats were used in these experiments, is it just one? These experiments should be repeated in different rats to increase N*.

We apologize for this lack of clarity. In the original manuscript we presented the data in terms of number of electrodes and cells, and not in number of rats. Altogether the data is from 12 different rats. Figure 7 presents the data for 2 individual electrodes in 2 different rats. All other panels in Figure 7 average the data from 12 electrodes recorded in 12 different rats (a single multi-contact electrode per rat). This is now described in the legend of Figure 7 of the revised manuscript.

*3) There is a clear and interesting difference of barrel and septal sensory responses, this should be followed up and confirmed with electrophysiological recording and identification of the recording sites*.

This is a very valid point, and we tried to address this point during the project, but eventually decided that the size of the septa is too small to be accurately targeted with our single unit recordings. In fact we feel that this point is one of the clear advantages of two photon imaging. More specifically in our electrophysiological experiments we used two methods to define the recording site. First, intrinsic imaging performed prior to electrode insertion as described in the Methods section (and Figure 1—figure supplement 2). With this method we were able to target the barrel, but not the septa. Second, at the end of experiments we replicated the location of the electrode by inserting a needle connected to a Hamilton syringe using the micromanipulator previously used to hold the electrode, and injecting dye at the recording site. We did not use electrical lesions as we reused our recording electrodes, and did not want to damage them by the current injection. In our opinion these methods were sufficiently reliable to determine that the electrodes were placed within the barrel boundaries (if close to the barrel center), but due to the small size of the septa relative to the barrels, these methods were not accurate enough to distinguish between septal and barrel perimeter locations.

4) The authors perform biocytin electroporation of imaged cells to locate the imaging site relative to the barrel. This is a nice technique and important for the conclusions of the paper but more analysis should be presented for a main figure. Is the yellow arrow in D an identified single electroporated cell? Are there confirmed neurons that have been imaged and stained: this should be presented? Was electroporation done in all experiments? Are there any axons visible going down through the barrel?

The yellow arrow was intended to point generally to the region from where we performed the imaging and electroporation but was not intended to point to any specific neuron. Following the comment we now point the arrow to one of the neurons that was actually electroporated. In all cases several imaged cells were electroporated at the end of the experiment (a new panel is added to Figure 2). Electroporation was performed in 7 out of the 12 maps. In the remaining 5 maps we determined the location of the imaging site relative to the barrel using alignment of vertical blood vessels. In addition in all cases intrinsic optical imaging was used to target the barrel. This is now added to the legend of Figure 6. As to the axonal track, we used only brief electroporation times just to mark the cell bodies and we processed the animal shortly after so, we could not follow axonal processes.

*5) The free hand line scan method the authors use scans an optimal path through a number of neurons to provide a fast scan rate. The authors should describe how stable the optimal path is throughout their experiment. Small movements of the brain could end up in scanning neuropil. So, was the path corrected during recording to account for brain movement? A full frame image at the start and end of recording showing any brain movement would be a good control and should be presented*.

Indeed this is an important issue which we were aware of, and performed the necessary corrections. As a rule full frame images were obtained every 3-5 line scan images and if needed corrections were performed. Generally as the animal was anaesthetized, and the head was held by a very stable platinum plate, movements were small and only minor corrections were needed. This point is described now in the text and Methods section and in a new supplementary figure (Figure 1—figure supplement 1).

Reviewer 2:

*[…] My main questions were about what to make of this finding relative to features we know about and that we know have spatial maps; is this a re-discovery of direction tuning, because the frictional term of a given surface dictates how a vibrissa is stopped with high deceleration* (*e.g., caudal direction tuning?), or propensity to spring forward suddenly* (*e.g., rostral direction tuning?)? Similarly, while 'coarseness' has not been shown to have a map, frequency has been shown to have one related to biomechanical resonance, a property engaged by contact with sandpapers*.

The reviewer raises an important general question of whether the mapping of coarseness we are reporting is in fact reflects mapping of previously reported qualities such as orientation tuning. In this project we decided to treat texture coarseness as a single wholesome quality due to its biophysical importance. However, we are well aware of the fact that texture coarseness is composed of individual physical properties, and in turn one or more of these properties might be mapped onto the barrel. Previous studies have shown that whisker deflection direction is mapped onto the barrel. Although this is an interesting possibility at this point it is hard for us to see how textures differ from each other by the whisker deflection direction they evoke. However, we cannot rule out the possibility that the texture mapping would be a composite of other maps such as direction and other qualities that are not yet known to be mapped. We added a comment in the Discussion of the manuscript.

With respect to mapping of frequencies in different barrels, see answer to comment 1.

Analyses and clarifications conducted with the present data:

1) Did barrel identity in its anterior-posterior position in the barrel map predict the coarseness tuning of cells in that column?

Following the reviewer's comment we plotted the dominant frequency preference as a function of the barrel location along the row and arc of the barrel field (Figure 10). We did not see any spatially consistent trend.Author response image 1.The relationship between barrel location and dominant texture preference. Each barrel was assigned a dominant texture, which was the texture preferred by the largest number of neurons recorded in the barrel. The number of barrels with the different dominant textures was plotted as a function of the row (A) and arc (B) of the barrel (blue P120, red P320, green P600 and yellow P1000).

*2) When looking at the normalized barrel map, there seemed to be a lower left* (*presumably aligned to the somatotopic more ventral/posterior position on the face) to upper right gradient of frequency preference. For example, no cells preferring 120 were outside of the lower left, and the majority of cells preferring the highest frequency are upper right. Smoothing these data lightly and integrating across frequencies should be conducted*.

As suggested by the reviewer we performed smoothening of the normalized barrel map (described in Methods section and Figure 6). As pointed out by the reviewer indeed the new smoothed maps, and especially the passive smoothed map, raise the possibility for a caudal–rostral/medio-lateral axis arrangement of coarseness texture in the barrel. We raised this possibility in the revised manuscript.

*3) Also, please plot the actual axes of orientation used to align the barrels on this summary image, and please make sure all maps are presented in this same orientation for consistency*.

Actual axes are added and orientations were double-checked.

*4) Is there a spatial radial or* (*as suggested in question 2) a lower left-upper right map for frequencies directly played into the vibrissae? These data should be plotted on a normalized barrel map also*.

*5) More generally, while I buy the argument that calcium signals cannot track individual stick-slip events, the mean frequency generated by artificial whisking on each sandpaper can be reported, as can the range and peak velocity values, as can the number of stick-slip events per grit. These properties for vibrissae chosen to be identity- and length-matched to the ones used for the initial data acquisition* (*as I assume those vibrissae are now long gone) should be used to make these measures. At least we can begin to try to understand what mapping may exist from lower level features that have been probed and the present findings*.

As suggested by the reviewer we further analyzed the data we acquired during our experiments when the whiskers contacted different sand papers. The revised manuscript now includes the power spectra (Figure 8—figure supplement 1), angle at base and curvature (Figure 1—figure supplement 2) and the properties of the stick-slip events for the different sand papers. The analysis is presented in a new supplementary figure (Figure 1—figure supplement 4).

6) The authors simply state that the contact point of the vibrissa was related to its 'not getting stuck'; can we have any more information? Was the contact point of the vibrissa on the surface of the sandpaper a typical distance from the face and/or from the vibrissa tip?

Typically the contact point from the tip of the vibrissa to the sand paper was set to just touching the sand paper as was judged using stereoscope (added to the Methods section).